# Directed Graph Transformers

**Qitong Wang**                                                    *wangq19@rpi.edu*
*Rensselaer Polytechnic Institute*

**Georgios Kollias**                                              *gkollias@us.ibm.com*
*IBM Research*

**Vasileios Kalantzis**                                          *vkal@ibm.com*
*IBM Research*

**Naoki Abe**                                                     *nabe@us.ibm.com*
*IBM Research*

**Mohammed J. Zaki**                                             *zaki@cs.rpi.edu*
*Rensselaer Polytechnic Institute*

**Reviewed on OpenReview:** *https://openreview.net/forum?id=otTFPjziiK*

## Abstract

In this paper, we address the problem of capturing graph directionality using transformers. Most existing graph transformers typically capture distances between graph nodes and do not take edge direction into account. This is a limiting assumption since many graph applications need to exploit sophisticated relationships in graph data, such as time, causality, or generic dependency constraints. We introduce a novel graph transformer architecture that explicitly takes into account the directionality between connected graph nodes. To achieve this, we make use of dual encodings to represent both potential roles, i.e., source or target, of each pair of vertices linked by a directed edge. These encodings are learned by leveraging the latent adjacency information extracted from a directional attention module, localized with $k$-hop neighborhood information. Extensive experiments on synthetic and real graph datasets show that our approach can have significant accuracy gains over previous graph transformer (GT) and graph neural network (GNN) approaches, providing state-of-the-art (SOTA) results on inherently directed graphs.

## 1 Introduction

Graphs are one of the most general and versatile data structures encountered in diverse application domains, ranging from biology and social networks to transportation and finance. Analyzing the graphs that arise from such applications and discovering patterns in them is of paramount importance in the associated domains. An important property of a graph is whether its edges are directed or not. Directed graphs are natural representations of relations including social connections, human communications, paper citations, financial transactions, Web links, and causes and effects. The state-of-the-art methods for analyzing directed graphs use Graph Neural Networks (GNNs) to learn node and directed edge encodings for tasks like link prediction (Kollias et al., 2022; Salha et al., 2019), node classification (Zhang et al., 2021) and graph-level tasks (Beaini et al., 2021).

In this paper, we address the relatively unexplored problem of analyzing *directed* graphs using graph transformers (GTs). Transformers hold the promise of enhanced performance over GNNs due to their ability to

represent entities without enforcing the inductive adjacency bias (Vaswani et al., 2017), and due to their *dynamic* multi-head attention mechanism, in contrast to GNNs where the attention is hardwired in *static* edge weights. This flexibility of GTs comes, however, with the challenge of modeling directed graph structures. Most existing GTs focus on integrating only the graph connectivity structure into the Transformer. They do not prioritize how to reflect the directionality of graph edges in their proposed architecture. This is either because some key techniques are not applicable to directed graphs (e.g., Laplacian eigenvectors (Dwivedi & Bresson, 2021)) or the edge-direction information is encoded as static, fixed scalars (either local in/out degrees in (Ying et al., 2021) or pairwise shortest path distances in (Hussain et al., 2022; Ying et al., 2021)).

We introduce *Directed Graph Transformer* (DiGT), a novel GT architecture that explicitly takes into account graph directionality. The crux of this architecture is that it incorporates both edge direction and graph connectivity structure into the standard Transformer architecture (Vaswani et al., 2017) as first-class citizens. Edge direction is represented by dual encodings for each graph node capturing its potential role as either a source or target of a directed edge. Source and target encodings are learned using a multi-head *directional attention* module that incorporates edge channels as bias. By interpreting attention matrices as latent adjacency matrices, our technique updates a node's source vector by aggregating the target vectors of the neighbors it points to, after incorporating suitable learnable parameters; similarly, a node's target vector update is the aggregation of the source vectors of those neighbors pointing to it.

Although the general idea of using dual encodings at each graph node has been explored in various non-GT methods for directed graph learning, its application to the domain of GTs is novel and comes with its challenges. The main difference is that existing methods leverage and rely on the existing directed graph structure: the dual encodings are statically computed based on graph properties or message passing occurs using the directed graph connectivity structure. This introduces a convolutional inductive bias: the neighborhood of a node only consists of the edges of the directed graph and cannot include latent edges which could be learned and produce more discriminative node encodings for directed graphs. In contrast, in DiGT dual node encodings are *dynamically* learned without using the explicit directed graph structure.

We evaluate our DiGT model architecture for directed graph classification tasks. A challenge is that publicly available directed graph benchmark datasets for this task are scarce and only capture directionality indirectly. For example, the directed graph dataset in (Hussain et al., 2022) is derived from `MNIST/CIFAR10` images which are inherently undirected. To this end, we introduce the `FlowGraph` family of directed graph datasets that explicitly relate the edge direction pattern in graphs to their classification labels. Using these new datasets, as well as existing `MNIST/CIFAR10` (Hussain et al., 2022), `Twitter` (Leskovec & Mcauley, 2012), and `Malnet-tiny` (Freitas et al., 2020) datasets, we empirically demonstrate that DiGT gives significant accuracy gains for the tasks on both synthetic and real graph datasets. Our extensive experiments reveal that when edge directionality is an inherent, rather than derivative, characteristic of the instances to be classified, DiGT provides state-of-the-art (SOTA) results, outperforming graph transformer based and GNN alternatives by a large margin.

## 2 Related work

**Methods for Directed Graph Learning.** Earlier works on analyzing directed graphs are based on matrix factorization techniques to learn node encodings, such as Singular Value Decomposition (SVD) of higher-order adjacency matrices exploring the directed $k$-hop neighborhood of a node (Ou et al., 2016), or Non-negative Matrix Factorization (NMF) (Sun et al., 2019). Another line of work focuses on analyzing special matrix forms of adjacency information such as the Hermitian adjacency matrix of the directed graph (Cucuringu et al., 2020) or learning linear combinations of powers of the directed graph adjacency matrix and its transpose (He et al., 2021). More recently, GNNs have been used (Kollias et al., 2022; Tong et al., 2020a;b; Zhang et al., 2021) that operate based on a message-passing architecture and provide higher learning flexibility due to the usage of learnable weight matrices that multiply the node encodings. Graph Attention Network (GAT) (Veličković et al., 2017) is a GNN that incorporates local self-attention resembling a transformer.

A limitation of all the aforementioned approaches is that they critically rely on the explicit directed graph structure (adjacency matrix): (a) The $k$-hop neighborhood learning techniques (He et al., 2021; Ou et al.,

2016; Sun et al., 2019; Zhou et al., 2017) involve matrix factorization or composition of powers of known adjacency matrices, or random walks over the graph structure. (b) Special matrix forms of adjacency information used in (Cucuringu et al., 2020; Tong et al., 2020a) allow directed graphs as input. (c) GNNs in (Kollias et al., 2022; Salha et al., 2019; Tong et al., 2020b; Veličković et al., 2017; Zhang et al., 2021) are message-passing models, and single or dual-node encoding messages can flow only through existing edges. Reliance on the directed graph structure introduces inductive bias during learning: latent edges that could positively contribute to the learning problem at hand can be missed as a result. In contrast, DiGT does not rely on the directed graph structure (in effect it assumes full graph connectivity) and learns the edge weights by exchanging the dual node encodings between the nodes.

We note here that the general idea of dual node encodings has also been used in several of the above works. However, such encodings are typically computed by exploiting the directed graph structure. Applying this idea in GTs is challenging because there are no assumptions on graph structure (i.e., assuming full connectivity) and they need to be learned in a dynamic manner.

**Graph Transformers** GTs were introduced in (Dwivedi & Bresson, 2021), which proposed two inspiring GT architecture variants. The first variant produces only node encodings, while the second variant is augmented to also produce edge encodings. Node encodings follow the standard Transformer architecture (Vaswani et al., 2017), while edge encodings are updated by scaling the attention matrix. They attend only to existing neighbors (local self-attention), so a strong inductive bias is enforced. SAN (Kreuzer et al., 2021) uses learned positional encodings (LPE) to enhance the learning of graph structure. SAT (Chen et al., 2022) enhances the learning by extracting $k$-hop subgraphs. In Graphormer (Ying et al., 2021), the attention aperture critically expands to all nodes (global self-attention). They propose adding and learning node encodings that are functions of input and output degree centralities (centrality encoding), and arbitrary node pairs are represented by two bias terms to the attention matrix (spatial and edge encodings). In EGT (Hussain et al., 2022), they combine ideas from (Dwivedi & Bresson, 2021) (separate channels for nodes and edges, scaling and gating the attention matrix) and from (Ying et al., 2021) (global self-attention, bias terms from spatial encoding, however, learned from the edge channels) to yield an effective GT approach. Recently, hybrid models that combine graph neural networks with graph transformers have been proposed, such as GraphGPS (Rampášek et al., 2022) and Exphormer (Shirzad et al., 2023), that attain competitive performance results, while aiming at scalability. GraphGPS is a framework for combining pluggable encodings, local message passing, and global attention modules; Exphormer introduces a sparse attention mechanism based on global virtual nodes and expander graphs. In Digraph Transformer (Digraph-T) (Geisler et al., 2023), they learn over directed graphs using Transformers, leveraging the eigenvectors of the Magnetic Laplacian matrix (Furutani et al., 2020) as position encodings, as well as directional random walks. Whereas we also use position encodings (PE), including the Magnetic Laplacian, it is only a minor component of our overall architecture. Our main contribution is the integration of bidirectional attention for graph transformers.

Our DiGT approach is a global self-attention transformer, learning both dual node encodings and edge encodings (dual-channel architecture). A node encoding in DiGT consists of a pair of source and target vectors that capture the edge direction semantics. Therefore, in downstream tasks that require directionality and take node encodings as input, DiGT provides embeddings of high discriminative power. In comparison, Graphormer (Ying et al., 2021), EGT (Hussain et al., 2022), and the other graph transformers (Dwivedi & Bresson, 2021; Zhang et al., 2020) produce only single-vector node embeddings that cannot differentiate the direction of an edge; and, as already mentioned, directed GNNs (Kollias et al., 2022; Salha et al., 2019; Tong et al., 2020b; Veličković et al., 2017; Zhang et al., 2021) produce dual node embeddings that suffer from convolutional inductive bias, that is restricted to only the given neighborhood structure.

## 3 Directed Graph Transformer Architecture

We now describe our directed graph transformer DiGT, whose main components are shown in Figure 1. DiGT uses three main ideas: dual node embeddings – for source and target representations, with a HITS-inspired (Kleinberg, 1999) aggregation combined with learnable implicit adjacency information via directed attention, as well as using $k$-hop neighborhood virtual edges. Table 1 shows the notations used below. Our model contains multiple layers and multiple attention heads, but we omit these for ease of presentation.

Table 1: Mathematical Notation

| Notation | Meaning | Shape | Notation | Meaning | Shape |
|---|---|---|---|---|---|
| $V$ | The set of graph nodes | | $E$ | The set of graph edges | |
| $n,e$ | Number of nodes or edges | | $d,d_e$ | Dimension of input node feature or edge feature | |
| $h$ | Number of heads | | $d_p$ | Dimension of attention layer. $d_p = d/h$ | |
| $\mathbb{A}$ | Adjacency matrix | $n \times n$ | $\mathbf{D}_{ST}^{(k)}, \mathbf{D}_{TS}^{(k)}$ | $k$-hop filter matrix from source-target or target-soruce | $n \times n \times 1$ |
| $\mathbf{S}, \mathbf{T}$ | Initial node encodings for source or target nodes | $n \times d$ | $\mathbf{E}_{ST}, \mathbf{E}_{TS}$ | Edge features for source-target | $n \times n \times h$ |
| $\mathbf{Q_S}, \mathbf{K_S}, \mathbf{V_S}$ | Query, Key, Value for source nodes | $n \times d_p$ | $\mathbf{G}_{ST}, \mathbf{G}_{TS}$ | Gate matrix for source-target or target-source | $n \times n \times h$ |
| $\mathbf{Q_T}, \mathbf{K_T}, \mathbf{V_T}$ | Query, Key, Value for target nodes | $n \times d_p$ | $\mathbf{A}_{ST}, \mathbf{A}_{TS}$ | Attention matrix before softmax for source-target or target-source | $n \times n \times h$ |
| $\mathbf{Y}$ | Value representation | $n \times d_p$ | $\tilde{\mathbf{A}}_{ST}, \tilde{\mathbf{A}}_{TS}$ | Attention matrix after softmax for source-target or target-source | $n \times n \times h$ |

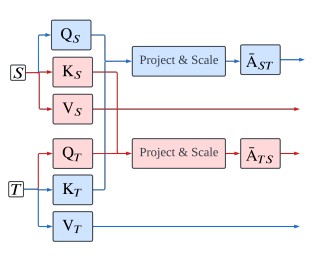
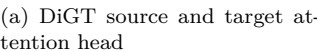
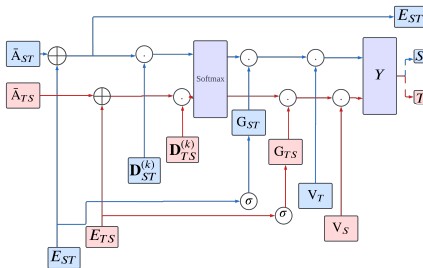

(a) DiGT source and target attention head

(b) DiGT layer structure

Figure 1: The two figures show the DiGT architecture. In (a), the attention is calculated in both directions for each node. Then, in (b), a softmax operation is applied on the stacking dimension to compute the importance of directionality (i.e., softmax is over the ST and TS axes). Blue nodes and arrows represent the flow of source embeddings, red nodes and arrows represent the flow of target embeddings, and purple nodes represent the operations on both source and target embeddings.

## 3.1 Input Layer

We represent a directed *graph* as $G(V, E)$; $V$ is the set of $n = |V|$ graph nodes, $E = \{(i, j) \in V \times V : i \mapsto j\}$ is the set of its $m = |E|$ directed edges. Each node $i$ is equipped with a pair of vectors in $\mathbb{R}^d$, $1 \leq i \leq n$: (i) vector $\mathbf{s}_i$ encodes $i$'s role as a source, which is the same for any of the directed edges it participates in as a source, and (ii) vector $\mathbf{t}_i$ encodes $i$'s role as a target.

Positional encodings play an important role in comprehending graph structures for GNNs. Traditional methods, such as the graph Laplacian, are limited to undirected graphs due to their requirements of a symmetric adjacency matrix. In our study, we select two techniques to overcome this limitation: Singular Value Decomposition (SVD), and the Magnetic Laplacian (Geisler et al., 2023). We detail these two positional encoding methods in the Appendix. For a given adjacency matrix $\mathbb{A}$, we denote by $\mathbf{P}_S$ and $\mathbf{P}_T$ the positional encodings for the source and target nodes, respectively. When input node features $\mathbf{N}_f$ are available (set $\mathbf{N}_f = 0$, otherwise), the input/initial *node* embeddings for the DiGT model are given as

$$\mathbf{S} = L_s(\mathbf{P}_S) + L_f(\mathbf{N}_f) \qquad\qquad \mathbf{T} = L_t(\mathbf{P}_T) + L_f(\mathbf{N}_f) \qquad\qquad (1)$$

where $L_s$, $L_t$ and $L_f$ are learnable linear transformations (subscripted as $s$ for the sources, $t$ for targets, and $f$ for input features), and $\mathbf{S}, \mathbf{T} \in \mathbb{R}^{n \times d}$. To encode the edges, considering input edge features $\mathbf{E}_f$ (set $\mathbf{E}_f = 0$ otherwise), the input/initial *edge* embeddings for the DiGT model are given as

$$\mathbf{E}_{ST} = L_e([\delta_{st}]_{s,t=1,...,n}) + L_{ef}(\mathbf{E}_f) \qquad\qquad (2)$$

where $L_e$ is an embedding layer, $L_{ef}$ is a learnable linear transformation, and $\delta_{st}$ is the shortest directed path distance from source $s$ to target $t$, clipped at maximum $k$-hops (if $t$ is not reachable from $s$, we set $\delta_{st} = k + 1$). The result, $\mathbf{E}_{ST} \in \mathbb{R}^{n \times n \times h}$, is the matrix of $h$ dimensional edge embeddings, and we set $\mathbf{E}_{TS}$ as the transpose of $\mathbf{E}_{ST}$ along the first two dimensions.

### 3.2 DiGT Attention Layer

**Inspiration:** Given the dual node encodings, we need to determine the relationship between the source and target encoding vectors of different nodes, which will be used for updates in our GT architecture. For this, we draw high-level inspiration from the HITS (Kleinberg, 1999) centrality algorithm that computes two scalar-valued *hub* and *authority* scores for each node in a directed graph – a source node with a high hub score refers to (or points to) target nodes that contribute relevant information (in our case, for learning), and thus gain elevated authority scores. Consider for the moment one-dimensional or scalar source and target node embeddings, $s_i$ and $t_i$, which serve as the hub and authority score, respectively; we can express their relationship as $s_i = \sum_{i \mapsto j} t_j$ (i.e., good hubs point to good authorities) and $t_i = \sum_{j \mapsto i} s_j$ (i.e., good authorities are pointed to by good hubs).

Generalizing to our $d$-dimensional source and target encoding *vectors* $\mathbf{s}_i$ and $\mathbf{t}_i$, we could analogously write $\mathbf{s}_i = \sum_j \mathbb{A}_{ij} \mathbf{t}_j$ and $\mathbf{t}_i = \sum_j \mathbb{A}_{ji} \mathbf{s}_j$, or more compactly as $\mathbf{S} = \mathbb{A}\mathbf{T}$ and $\mathbf{T} = \mathbb{A}^\top \mathbf{S}$. Conceptually, $\mathbf{s}_i$ and $\mathbf{t}_i$ play the role of multi-dimensional hub and authority scores.

**Implicit and Directed Adjacency via Attention:** The key insight in DiGT is that we should not rely on the fixed adjacency matrix $\mathbb{A}$; rather, we should construct an *implicit* adjacency matrix, denoted $\bar{\mathbf{A}}$, by exploiting the attention mechanism. A straightforward approach to compute $\bar{\mathbf{A}}$ could be $\bar{\mathbf{A}} = \mathbf{S}\mathbf{T}^\top$. However, we need to make this learnable. To allow the flexibility of learning weight matrices for computing the implicit adjacency we use dual attention mechanisms. For the source nodes $\mathbf{S}$, let

$$\mathbf{Q}_S = \mathbf{S}\,\mathbf{W}_{QS} \qquad\qquad \mathbf{K}_S = \mathbf{S}\,\mathbf{W}_{KS} \qquad\qquad \mathbf{V}_S = \mathbf{S}\,\mathbf{W}_{VS} \qquad (3)$$

and similarly for the target nodes $\mathbf{T}$, let

$$\mathbf{Q}_T = \mathbf{T}\,\mathbf{W}_{QT} \qquad\qquad \mathbf{K}_T = \mathbf{T}\,\mathbf{W}_{KT} \qquad\qquad \mathbf{V}_T = \mathbf{T}\,\mathbf{W}_{VT} \qquad (4)$$

where all $\mathbf{W}$'s $\in \mathbb{R}^{d \times d_p}$ are learnable weight matrices, and $d_p$ is the projection dimensionality (suitably scaled down). As shown in Figure 1a, we obtain a pair of attention matrices

$$\bar{\mathbf{A}}_{ST} = \left(\mathbf{Q}_S \mathbf{K}_T{}^\top\right)/\sqrt{d_p} \qquad\qquad \bar{\mathbf{A}}_{TS} = \left(\mathbf{Q}_T \mathbf{K}_S{}^\top\right)/\sqrt{d_p} \qquad (5)$$

That is, the attention matrix $\bar{\mathbf{A}}_{ST}$ treats the source nodes as queries and the target as keys to compute their similarity, and vice-versa for $\bar{\mathbf{A}}_{TS}$ (see Figure 1a).

**Edge Feature and Neighborhood Attention:** We now allow for the edge channels to directly influence the attention by introducing the edge feature matrix, $\mathbf{E}_{ST} \in \mathbb{R}^{n \times n \times h}$, and *gate* matrix, $\mathbf{G}_{ST} \in \mathbb{R}^{n \times n \times h}$, which is linear transformations from $\mathbf{E}_{ST}$ (with added layer norms). Further, $\mathbf{E}_{TS}$ and $\mathbf{G}_{TS}$ are their transpose matrices, respectively.

Next, we *localize* the attention from node channels to the $k$-hop neighborhood around each node. This is implemented by masking the attention matrix along with the edge bias via an element-wise product with the binary $k$-*hop filter matrix*, defined as $\mathbf{D}_{i,j}^{(k)} = \{1 \ \textit{iff} \ \delta_{ij} \leq k, 0 \ \textit{iff} \ \delta_{ij} > k\}$, where $\delta_{ij}$ denotes the shortest path distance from node $i$ to node $j$. Thus, the attention matrices, denoted $\tilde{\mathbf{A}}$, for this layer are given as:

$$\tilde{\mathbf{A}}_{ST} = \left(\bar{\mathbf{A}}_{ST} + \mathbf{E}_{ST}\right) \odot \mathbf{D}_{ST}^{(k)} \qquad\qquad \tilde{\mathbf{A}}_{TS} = \left(\bar{\mathbf{A}}_{TS} + \mathbf{E}_{TS}\right) \odot \mathbf{D}_{TS}^{(k)} \qquad (6)$$

where $\mathbf{D}_{ST}^{(k)}$ and $\mathbf{D}_{TS}^{(k)}$ are the filter matrices in the two directions. This way we control the attention to be within the $k$-hop neighbors around each node.

**Directional Attention:** For any two nodes within a graph, our objective is to determine the direction of message passing between them. To achieve this, we compare the attention values in both directions. The larger of these values will indicate the predominant direction of message passing. Therefore, unlike traditional transformers that compute node importance via a softmax along each *row* of the attention matrix,

we stack both $\tilde{\mathbf{A}}_{ST}$ and $\tilde{\mathbf{A}}_{TS}$ and compute the softmax along the stacking direction (Figure 1b visualizes this mechanism), given as

$$\tilde{\mathbf{A}}_{ST}, \tilde{\mathbf{A}}_{TS} = \texttt{softmax}(\tilde{\mathbf{A}}_{ST}, \tilde{\mathbf{A}}_{TS}). \tag{7}$$

Note that directional attention combined with the $k$-hop filter matrix $\mathbf{D}^{(k)}$ applies a *soft-thresholding* for unidirectional edges, as opposed to the hard-thresholding that is typically used when masking with $-\infty$. That is, if we were to set $\mathbf{D}^{(k)}_{i,j} = -\infty$ if $\delta_{ij} > k$, then information would flow only along the direction that exists, and it would ignore the opposite direction for unidirectional edges. We show conclusively in our ablation study in Section 4.3 that soft-thresholding yields better performance.

Finally, we enable the flow of information between nodes by gating their value representations prior to aggregation; this is realized as multiplication by the sigmoid function, $\sigma()$, of the entries in *gate* matrices, $\mathbf{G}_{ST}$ and $\mathbf{G}_{TS}$, resulting in

$$\mathbf{Y} = \left((\tilde{\mathbf{A}}_{ST} \odot \sigma(\mathbf{G}_{ST}))\, \mathbf{V}_T\right) + \left((\tilde{\mathbf{A}}_{TS} \odot \sigma(\mathbf{G}_{TS}))\, \mathbf{V}_S\right), \tag{8}$$

where $\mathbf{Y} \in \mathbb{R}^{n \times d_p}$ is the value representation for one head. So, when we have $h = d/d_p$ heads, we concatenate all of them (and add layer norm) to obtain the final value representation $\mathbf{Y} \in \mathbb{R}^{n \times d}$, for the next step. Also, the different DiGT layers do not share edge embeddings and this is also true for bias and gate matrices.

### 3.3 Output Layers and Prediction

After each DiGT layer, we take the combined value encoding $\mathbf{Y}$, and use layer normalization and feed-forward network modules with residual connections to produce the node and edge encoding outputs for a DiGT layer. These outputs become inputs for the next layer. Thus, the updated dual encodings $\mathbf{S}'$, $\mathbf{T}'$ are given as:

$$\mathbf{S}' = f(L_{YS}(\mathbf{Y})) \qquad\qquad \mathbf{T}' = f(L_{YT}(\mathbf{Y})), \tag{9}$$

where, $L_{YS}$ and $L_{YT}$ are two linear transformations followed by a non-linear activation $f$ (with layer norms and residual connections). To obtain the updated edge embeddings $\mathbf{E}'_{ST} \in \mathbb{R}^{n \times n \times h}$ for the next layer, we apply a similar function on $\bar{\mathbf{A}}_{ST}$ as follows:

$$\mathbf{E}'_{ST} = f(L_E(\bar{\mathbf{A}}_{ST})) \tag{10}$$

Lastly, after the last DiGT layer is processed, the encodings $\mathbf{X} = \texttt{concat}(\mathbf{S}', \mathbf{T}')$ are driven through some final task-specific learning modules. These are typically multilayer perceptron layers (MLP) for tasks related to *node* and *edge* learning (node classification, link prediction), or pooling layers for *graph-level* learning (graph classification, graph regression). For the directed graph classification task, we use *global average pooling* as our main method for producing a representation/encoding of the whole graph; this is essentially the average of the final node encodings. We also experiment with the method of *virtual nodes* based pooling (Hussain et al., 2022): a clique of artificial nodes (virtual nodes) are added to each graph and connected to all its nodes (with the edge directed only from a graph node to the virtual nodes). After training, we average the concatenated source and target node embeddings of the virtual nodes and leverage the same final MLP layers for the downstream task. We examine the effects of these choices in the ablation studies.

## 4 Experiments

Our experiments were performed on NVIDIA V100 GPUs, with 32GB memory, using PyTorch. We provide additional experimental details in the Appendix.

### 4.1 Directed Graph Datasets

There are several directed datasets used in previous studies, such as MNIST (LeCun & Cortes, 2005), CIFAR10 (Krizhevsky et al., 2009), Ogbg-Code2 (Hu et al., 2020), and Malnet-tiny (Freitas et al., 2020). See the Appendix for dataset statistics.

Table 2: Randomized directionality via edge flips: Model performance

| | MNIST | CIFAR10 | Model | Ogbg-Code2 | Model | Malnet-tiny | Malnet-sub |
|---|---|---|---|---|---|---|---|
| EGT | 98.17 +/- 0.09 | 68.70 +/- 0.41 | DAG | 20.2 +/- 0.2 | Exphormer | 94.02 +/- 0.21 | 79.71 +/- 0.37 |
| EGT-Flip50 | 97.99 +/- 0.09 | 67.28 +/- 0.38 | DAG-Flip50 | 19.0 +/- 0.1 | Exphormer-Flip50 | 87.90 +/- 1.65 | 71.51 +/- 0.45 |

| Model | FlowGraph2 | FlowGraph3 | FlowGraph6 | Twitter3 | Twitter5 |
|---|---|---|---|---|---|
| DiGT | 98.00 +/- 0.54 | 74.61 +/- 1.95 | 46.03 +/- 0.41 | 93.33 +/- 0.64 | 86.67 +/- 0.52 |
| DiGT-Flip50 | 49.67 +/- 1.39 | 32.33 +/- 1.66 | 16.78 +/- 0.04 | 82.96 +/- 1.13 | 65.44 +/- 0.38 |

**MNIST and CIFAR10:** (Hussain et al., 2022) used MNIST and CIFAR10 as collections of directed graph inputs for their graph classification task, following the introduction of these datasets in (Dwivedi et al., 2020). They are originally collections of images (respectively, of handwritten digits and objects) and not graphs. In (Dwivedi et al., 2020), they convert an image to a directed graph by first segmenting the original image pixels into sets of (coarser-grained) SLIC superpixels (Achanta et al., 2012). Directionality in the edges then follows from the fact that if a superpixel $i$ has *fewer* neighboring superpixels than one of its neighbors $j$, it will tend to connect to $j$ rather than the other way around (i.e., $i \mapsto j$ will have more weight than $j \mapsto i$).

**Malnet-tiny and Malnet-sub:** Malnet-tiny provides a graph classification task for five different types of malicious software. It contains 5,000 graphs, and each graph contains less than 5,000 nodes. Its graph size is an obstacle for many graph transformers. We apply a filter to generate the Malnet-sub dataset: we choose the graphs with fewer than 500 nodes for training sets, and those with fewer than 2,000 nodes for both validation and test datasets. As a result, the Malnet-sub dataset contains 2,444 graphs. Malnet-sub dataset is more challenging than Malnet-tiny since it forces the models to effectively generalize from smaller graphs in the training data to larger, unseen graphs in the validation and testing phases[1].

**Ogbg-Code2:** The Ogbg-code2 dataset is a collection of Abstract Syntax Trees (ASTs) from thousands of Python method definitions extracted from 13,587 different popular repositories on GitHub. Given the AST and node features, the goal is code summarization, i.e., to predict the sub-tokens forming the method name.

### 4.1.1 Importance of Direction: Random Flip Test

Even though the above datasets are purported to be directed, our analysis points out a severe limitation, namely, direction plays little to no role in these datasets, unfortunately.

To evaluate the role of directionality in these datasets, we designed a simple random flip test. In essence, for a given edge flip probability, say $\theta$ (e.g., $\theta \in \{0.25, 0.5\}$), we flip each edge $(u, v)$ with probability $\theta$ during each of the training, validation and testing steps. If, even after randomizing the edge direction for $\theta$ fraction of the edges, a model consistently achieves accuracy comparable to that on the original dataset, this provides strong support that directionality is not a crucial factor for that dataset.

Table 2 shows the results with $\theta = 0.5$ (i.e., 50% of the edges are flipped). We use EGT (Hussain et al., 2022) on the MNIST and CIFAR10 graphs, Exphormer (Shirzad et al., 2023), which is the SOTA on Malnet-tiny, and DAGformer (Luo, 2022), which is the top-performer on the Ogbg-Code2 leaderboard (Hu et al., 2020). We observe that despite altering the direction of almost half of the edges on MNIST, CIFAR10 and Ogbg-Code2, the results remain largely unaffected: predictive performance drops at most 0.42% for MNIST, 1.42% for CIFAR10, and 1.2% for Ogbg-Code2 in absolute terms. These tiny differences indicate that directionality is not important for those datasets. On the other hand, there is a significant performance loss on Malnet-tiny and Malnet-sub indicating the importance of directionality for these. See the Appendix for results on different $\theta$ values and other models, too.

---

[1]For example, the current state-of-the-art model, Exphormer (Shirzad et al., 2023), achieves a high accuracy of 93.38% on Malnet-tiny, while its performance drops to 79.71% accuracy on Malnet-sub when constrained to 100,000 parameters.

### 4.1.2 Inherently Directed Datasets

Having shown that existing datasets used to evaluate directed graph transformers violate their *raison d'être*, we propose two classes of inherently directed graphs (see Appendix for details of dataset generation).

**FlowGraph datasets:** We introduce a family of directed graph datasets that explicitly relate the edge direction pattern in graphs to their classification labels. In each graph, we create multiple clusters of nodes interconnected by random edges and arrange these clusters in a sequential order. For directed edges linking nodes in consecutive clusters, namely cluster $l$ and cluster $l+1$, we define a directional flow: a set percentage $f\%$ of edges flow from cluster $l$ to $l+1$, while the remaining $1-f\%$ edges flow in the reverse direction. These directional flow percentages vary across different classes. We generate three distinct graph datasets with $2, 3$, and $6$ classes. Table 2 reveals that directionality is important in these datasets: `FlowGraph2` experiences a 48.33% drop in performance when 50% of the edges are reversed. Large drops are observed for the other FlowGraph datasets, too.

**Twitter datasets:** We use 973 directed ego-networks from Twitter[2], each corresponding to some user $u$ (*ego*): the ego-network is between $u$'s friends also referred to as *alters* (Leskovec & Mcauley, 2012). If nodes $v_i, v_j$ are in $u$'s ego-network then $u$ follows them and if $v_i$ follows $v_j$ then there is a directed edge $v_i \mapsto v_j$ in the ego-network. We introduce perturbations to each of these real ego-networks where a perturbation can be either (i) *rewiring* of an existing edge, or (ii) *reversing* of the direction of an existing edge. Leveraging these perturbations, we create two distinct graph datasets: `Twitter3` with 3 classes and `Twitter5` with 5 classes. These classifications are based on varying percentages of perturbed edges. In Table 2, we observe significant drops in performance with 50% edge flipping: `Twitter3` shows an absolute drop of 9.37%, whereas `Twitter5` shows a drop of 21.23%, indicating that directionality plays an important role for these datasets.

### 4.1.3 Degree of Directionality

To further cement the role of directionality for a dataset, we propose another measure – the *degree of directionality*. Let SCC denote a strongly connected component in a directed graph (a maximal subset of mutually reachable nodes). Further, given the set of $m$ SCCs of a directed graph, $\mathcal{S} = \{S_1, S_2, ..., S_m\}$, define the *SCC entropy* of the graph as follows: $E(\mathcal{S}) = -\sum_{i=1}^{m} p_i \log p_i$, where $p_i = |S_i|/n$, i.e., the distribution of

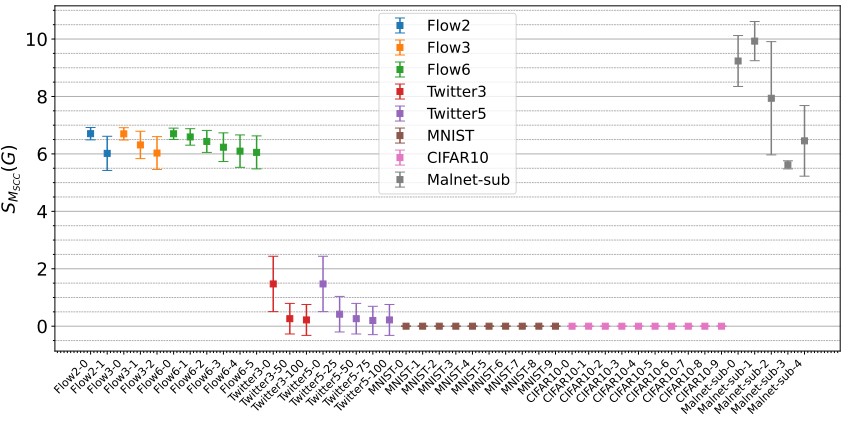

Figure 2: SCC entropy for each class for each dataset

SCC sizes. A low entropy means that most nodes are mutually reachable, and thus directionality is not expected to play a big role. On the other hand, larger SCC entropy values, with a maximum value of $\log n$, indicate smaller reachable components, which means that directionality is clearly important.

Figure 2 plots the SCC entropy for the different datasets; for each class, we plot the average and standard deviation[3]. We can see a very clear trend. `FlowGraph` classes are inherently directed, with larger entropies, and `Twitter` captures the inherent directionality of the "follow" relationship between entities, exhibiting lower entropy values. More importantly, the derived `MNIST` and `CIFAR10` have no inherent directedness, and their entropy values are extremely low. We will show that DiGT performs even better when direction matters, such as for `FlowGraph`, `Twitter`, and `Malnet-sub`.

---

[2]`https://snap.stanford.edu/data/ego-Twitter.html`
[3]We do not consider `ogb-code2` further, since directionality is not important for this dataset, and the task is code summarization instead of classification.

Table 3: *Classification accuracy of* DiGT *against various GNNs and GTs.*

| Model | FlowGraph2 | FlowGraph3 | FlowGraph6 | Twitter3 | Twitter5 | Malnet-sub | MNIST | CIFAR10 |
|---|---|---|---|---|---|---|---|---|
| GCN (Kipf & Welling, 2016) | 87.50 +/- 1.27 | 58.28 +/- 0.88 | 30.36 +/- 0.55 | 76.24 +/- 0.56 | 61.23 +/- 1.67 | 78.75 +/- 0.23 | 90.71 +/- 0.22 | 55.71 +/- 0.38 |
| GAT (Veličković et al., 2017) | 84.92 +/- 1.90 | 58.83 +/- 1.47 | 30.31 +/- 0.28 | 74.59 +/- 1.59 | 56.79 +/- 0.05 | 79.58 +/- 1.65 | 95.54 +/- 0.21 | 64.22 +/- 0.46 |
| GatedGCN (Bresson & Laurent, 2017) | 94.33 +/- 0.24 | 68.61 +/- 0.61 | 35.22 +/- 1.21 | 87.49 +/- 0.85 | 65.75 +/- 1.14 | 80.50 +/- 1.74 | 97.34 +/- 0.14 | 67.31 +/- 0.31 |
| GraphSage (Hamilton et al., 2017) | 92.92 +/- 0.43 | 65.55 +/- 0.21 | 34.14 +/- 0.61 | 70.28 +/- 0.43 | 57.20 +/- 1.02 | 76.77 +/- 1.73 | 97.31 +/- 0.10 | 65.77 +/- 0.31 |
| PNA (Corso et al., 2020) | 96.17 +/- 0.31 | 72.94 +/- 0.64 | 41.42 +/- 1.32 | 88.26 +/- 1.16 | 70.94 +/- 2.01 | 78.85 +/- 1.01 | 97.41 +/- 0.16 | 70.21 +/- 0.15 |
| GT (Dwivedi & Bresson, 2021) | 93.17 +/- 0.82 | 66.17 +/- 0.60 | 36.20 +/- 1.12 | 90.66 +/- 0.35 | 79.55 +/- 0.68 | 75.04 +/- 0.61 | 97.75 +/- 0.12 | 68.02 +/- 0.16 |
| SAN (Kreuzer et al., 2021) | 91.73 +/- 1.84 | 63.87 +/- 0.66 | 34.57 +/- 0.45 | 85.33 +/- 0.78 | 63.13 +/- 1.65 | 79.27 +/- 0.23 | 96.82 +/- 0.13 | 66.96 +/- 0.39 |
| Digraph-T (Geisler et al., 2023) | 95.42 +/- 0.82 | 72.39 +/- 0.32 | 40.81 +/- 0.76 | 87.24 +/- 0.32 | 70.77 +/- 1.94 | 73.28 +/- 1.52 | 96.06 +/- 0.03 | 65.35 +/- 0.45 |
| EGT (Hussain et al., 2022) | 95.00 +/- 1.67 | 72.06 +/- 1.16 | 42.97 +/- 0.62 | 86.49 +/- 0.73 | 73.94 +/- 1.47 | 72.35 +/- 1.38 | 98.17 +/- 0.09 | 68.70 +/- 0.41 |
| Exphormer (Shirzad et al., 2023) | 96.72 +/- 0.44 | 72.81 +/- 0.38 | 41.70 +/- 0.39 | 89.76 +/- 0.30 | 72.72 +/- 1.40 | 79.71 +/- 0.37 | **98.55 +/- 0.04** | **74.69 +/- 0.13** |
| DiGT | **98.00 +/- 0.54** | **74.61 +/- 1.95** | **46.03 +/- 0.41** | **93.33 +/- 0.64** | **86.67 +/- 0.52** | **80.78 +/- 1.76** | 98.02 +/- 0.10 | 67.05 +/- 0.20 |

## 4.2 Experimental Comparison

We compare DiGT with GCN (Kipf & Welling, 2016), GAT (Veličković et al., 2017), GatedGCN (Bresson & Laurent, 2017),GraphSage (Hamilton et al., 2017), and PNA (Corso et al., 2020) as the representatives of GNNs, and with GT (Dwivedi & Bresson, 2021), SAN (Kreuzer et al., 2021), EGT (Hussain et al., 2022), Exphormer (Shirzad et al., 2023), and Digraph-T (Geisler et al., 2023) as the representatives of GTs. We train all the models on `FlowGraph`, `Twitter`, `Malnet-sub`, and `MNIST/CIFAR10` respectively, restricted to 100K parameters. The accuracy results are listed in Table 3. Additional details can be found in the Appendix. Our source code is available via github: https://github.com/Qitong-Wang/Directed-Graph-Transformers.

For `MNIST` and `CIFAR10`, which are not intrinsically directed by construction, DiGT maintains competitive performance over GT and EGT, experiencing only a minor decrease in accuracy. On the other hand, we observe that for all the datasets where direction is important, DiGT outperforms existing models providing SOTA results. For instance, on the `FlowGraph6` dataset, DiGT outperforms the next best model EGT by 3.06% in accuracy; the performance gains against GNN alternatives are even larger. Likewise, for the `Twitter5` dataset, DiGT outperforms the next best model, GT, by over 7.12% in accuracy. Among the GNNs, PNA is the best but lags behind DiGT by a huge margin. For `Malnet-sub`, models that are adept at capturing local information, such as GCN and Exphormer, show good performance. In contrast, traditional transformers that focus on the whole graph-level attention, like GT and EGT face more challenges for this task. However, DiGT surpasses the next best model, Exphormer, by 1.07% in accuracy. These results indicate that the dual attention mechanisms in DiGT can effectively learn both local and graph-level information.

## 4.3 Ablation Studies

We present ablation studies demonstrating the effectiveness of incorporating dual-vector attention matrices and $k$-hop virtual edge filters. Additionally, further ablation studies regarding other architectural components of our models, such as positional encodings and virtual nodes, are detailed in the Appendix.

### 4.3.1 Single-Vector Node Embeddings

To study the benefit of our dual node encodings, we constrain DiGT to leverage only one vector embedding per node, and refer to this version as DiGT-OneEmb. This kind of ablation makes DiGT similar to EGT, so the expectation is to get accuracy values closer to EGT. Results in Table 4 confirm this intuition. On average, the *ablated* DiGT model scores between the reported values for DiGT (higher) and EGT (lower) and much closer to those of EGT, especially in the case of `Twitter` datasets, e.g., $86.04 \pm 0.45$ versus $86.49 \pm 0.73$ in EGT for `Twitter3`.

Table 4: *Ablation: Single-Vector Node Embeddings.*

| Model | FlowGraph2 | FlowGraph3 | FlowGraph6 | Twitter3 | Twitter5 | Malnet-sub | MNIST | CIFAR10 |
|---|---|---|---|---|---|---|---|---|
| DiGT | **98.00 +/- 0.54** | **74.61 +/- 1.95** | **46.03 +/- 0.41** | **93.33 +/- 0.64** | **86.67 +/- 0.52** | **80.78 +/- 1.76** | 98.02 +/- 0.10 | 67.05 +/- 0.20 |
| DiGT-OneEmb | 96.75 +/- 0.41 | 73.67 +/- 0.85 | 43.81 +/- 0.42 | 87.52 +/- 0.50 | 73.09 +/- 0.59 | 70.07 +/- 1.30 | 97.49 +/- 0.72 | 66.78 +/- 0.90 |
| EGT (Hussain et al., 2022) | 95.00 +/- 1.67 | 72.06 +/- 1.16 | 42.97 +/- 0.62 | 86.49 +/- 0.73 | 73.94 +/- 1.47 | 72.35 +/- 1.38 | **98.17 +/- 0.09** | **68.70 +/- 0.41** |

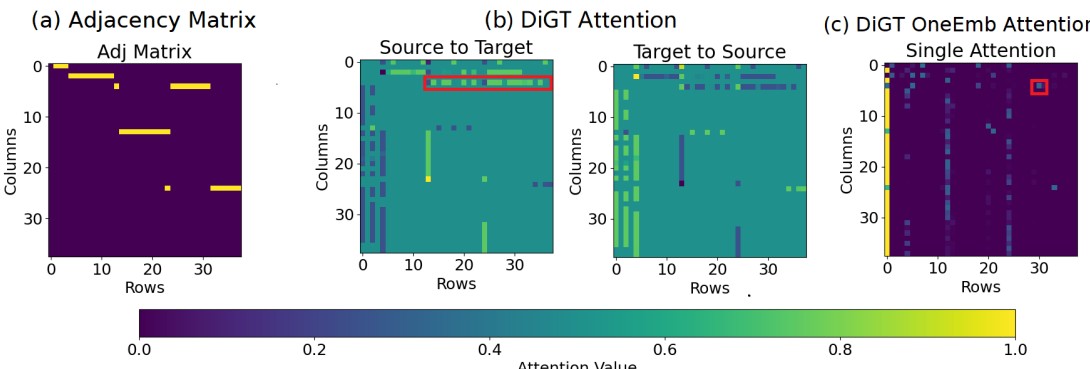

Figure 3: Visualization of adjacency and attention matrix for DiGT, for a selected `Malnet-sub` graph.

Additionally, Figure 3 visually represents attention matrices for a sample graph within the `Malnet-sub` dataset. Figure 3(b) provides a pair of attention matrices in DiGT: the left figure shows the attention directed from the source to the target, namely $\tilde{\mathbf{A}}_{ST}$ in Equation 8, and the right one depicts the attention flow from the target to the source, namely $\tilde{\mathbf{A}}_{TS}$. Figure 3(c) shows the attention for the DiGT-OneEmb variant. For all the figures, we sum up the last layer of the attention matrices from all heads and normalize the values from 0 to 1. Brighter colors correspond to higher attention values, indicating greater importance of the information at that specific location. In summary, these figures visualize the status of the attention matrices, which are ready to be multiplied (after gating) by the values $\mathbf{V}_T, \mathbf{V}_S$.

Traditional transformers struggle to discern critical patterns within a graph. This challenge is shown in Figure 3(c), where the most informative neighboring node for any given source node is indicated by the brightest node in each row. For instance, the red rectangle highlights one such pair. However, this does not necessarily imply that these two nodes are pivotal for the entirety of the figure. On the contrary, in Figure 3(b), the presence of a bright flow within the red rectangle suggests that all these nodes are crucial for the overall graph structure. The application of softmax across the stack direction of the two attention matrices enables the model to discern the relationships between each edge and its corresponding reversed virtual edge within $k$-hops, and the information flow comes from the union of the original graph and its reversed counterpart.

### 4.3.2 $k$-Hop Virtual Edge Filters

Graph transformers, operating under the assumption that all nodes are interconnected, can be interpreted as adding virtual edges among all nodes. We introduce a hyperparameter $k$ to constrain the addition of virtual edges, which has two critical functions: First, it sets the boundary for the shortest directed path distance between two nodes in Equation 2, which contributes to the edge features $\mathbf{E}_{ST}, \mathbf{E}_{TS}$ and gates $\mathbf{G}_{ST}, \mathbf{G}_{TS}$. Second, it determines the $\mathbf{D}_{ST}^{(k)}, \mathbf{D}_{TS}^{(k)}$ virtual edge filters in Equation 6, which restrict message passing among distant nodes. Above, we demonstrated that DiGT effectively captures graph patterns and identifies key node pairs. We now present an ablation study to explore the impact of varying $k$ on the model's performance, where 'unlimited' refers to setting $k = 25$ as a sufficiently large number for the edge embedding layer $L_e$ and not multiplying with $\mathbf{D}_{ST}^{(k)}, \mathbf{D}_{TS}^{(k)}$ filters in Equation 6.

Table 5: *Ablation: Number of Hops.*

| Model | FlowGraph2 | FlowGraph3 | FlowGraph6 | Twitter3 | Twitter5 | Malnet-sub | MNIST | CIFAR10 |
|---|---|---|---|---|---|---|---|---|
| DiGT | 98.00 +/- 0.54 | **74.61 +/- 1.95** | **46.03 +/- 0.41** | 93.33 +/- 0.64 | **86.67 +/- 0.52** | 80.78 +/- 1.76 | **98.02 +/- 0.10** | **67.05 +/- 0.20** |
| DiGT-hop 1 | **98.58 +/- 0.24** | 74.39 +/- 1.10 | 43.92 +/- 1.03 | **93.90 +/- 0.08** | 85.97 +/- 1.35 | 80.53 +/- 0.87 | 97.23 +/- 0.28 | 64.50 +/- 0.84 |
| DiGT-hop 3 | 98.00 +/- 0.54 | **74.61 +/- 1.95** | **46.03 +/- 0.41** | 93.33 +/- 0.64 | **86.67 +/- 0.52** | 80.09 +/- 0.95 | **98.02 +/- 0.10** | **67.05 +/- 0.20** |
| DiGT-hop 5 | 97.00 +/- 0.71 | 74.11 +/- 0.98 | 45.53 +/- 1.27 | 92.08 +/- 0.56 | 86.56 +/- 0.25 | 80.58 +/- 1.15 | 96.62 +/- 0.48 | 65.16 +/- 0.94 |
| DiGT-hop 12 | 96.42 +/- 0.42 | 70.28 +/- 1.59 | 44.72 +/- 0.48 | 92.42 +/- 0.66 | 85.85 +/- 0.42 | **80.78 +/- 1.76** | 97.12 +/- 0.22 | 66.40 +/- 0.12 |
| DiGT-hop unlimited | 96.00 +/- 0.71 | 71.61 +/- 1.17 | 44.75 +/- 1.37 | 92.59 +/- 0.58 | 86.60 +/- 1.69 | 78.31 +/- 0.74 | 96.62 +/- 0.45 | 65.00 +/- 0.10 |

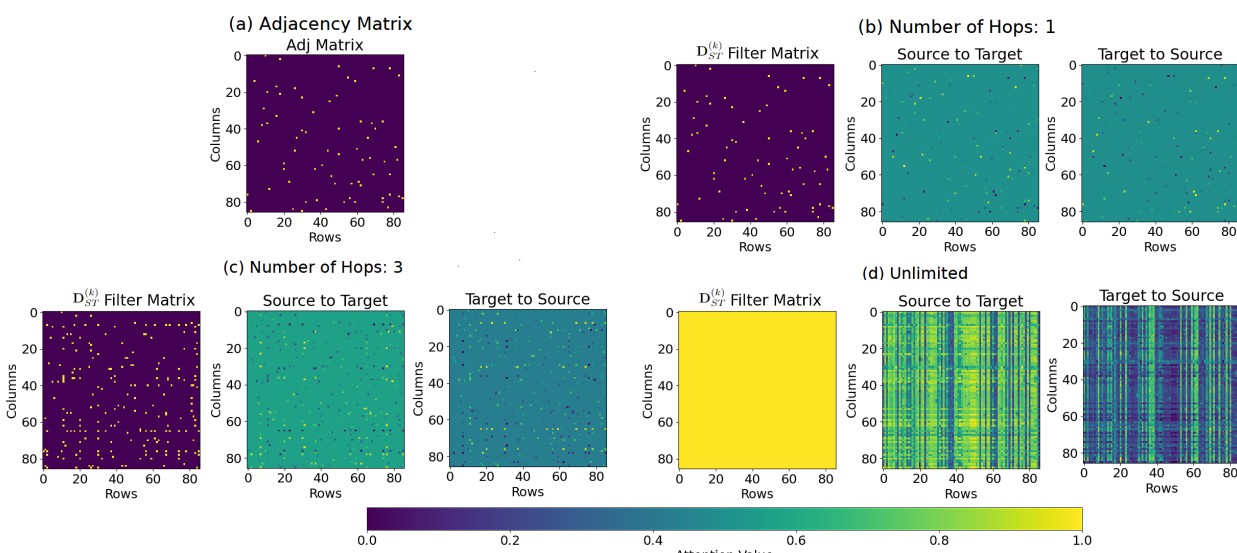

Figure 4: DiGT-attention matrices for a selected `FlowGraph6` graph.

Figure 4 illustrates that an increase in $k$ leads to more connections among nodes. In Figure 4(b), with $k = 1$, the model behaves similarly to a GNN. It has the ability to capture relationships beyond the adjacency matrix by using additional layers, but it is less effective than with higher $k$ values. Figure 4(d) illustrates a scenario with no hop limitations, allowing unrestricted message passing among all nodes. We can see that the model is capable of reconstructing the graph structure even under the assumption that all nodes are connected. However, the comparison in Table 5 illustrates that DiGT-hop unlimited does not achieve state-of-the-art performance on any datasets. On the other hand, the integration of filter $\mathbf{D}^{(k)}$ matrices, which inform the model about the graph's structure, can effectively improve the model's performance. As seen in Figure 4(c), when $k = 3$, the filter matrices enhance the connectivity of the original graph. Consequently, the attention matrix more accurately reflects important node relationships from the original graph and facilitates effective message passing between distant nodes. This configuration achieves the highest accuracy score on the `FlowGraph6` dataset.

Table 5 further indicates that the optimal number of hops varies with different datasets. For the `Malnet-sub` dataset, the highest accuracy is achieved at $k = 12$, with a significant drop when there are no hop limitations. This can be attributed to the large size of Malnet-sub graphs, which can have up to 2000 nodes, and their edges are sparse. As a result, a model requires a larger $k$ compared to smaller datasets. In contrast, we choose $k = 3$ for other datasets since they only contain approximately 200 nodes per graph. For datasets like `FlowGraph2` and `Twitter3`, $k = 1$ seems more effective, though these results still fall within one standard deviation compared to $k = 3$. In our experiments, we choose $k = 12$ for the `Malnet-sub` dataset and $k = 3$ for other datasets, as the default value.

### 4.3.3 Hard vs. Soft Thresholding for $\mathbf{D}^{(k)}$

Our $k$-hop virtual filter matrix $\mathbf{D}^{(k)}$ has been defined as $\mathbf{D}^{(k)}_{i,j} = \{1 \; iff \; \delta_{ij} \leq k, 0 \; iff \; \delta_{ij} > k\}$ where $\delta_{ij}$ denotes the shortest path distance from node $i$ to node $j$. In this case, 0 means there are no existing paths among two nodes. In this ablation study, we turn our focus to a variant definition, where $\mathbf{D}^{(k)}_{i,j} = \{1 \; iff \; \delta_{ij} \leq k, -\infty \; iff \; \delta_{ij} > k\}$ to examine how the performance is affected when we set an extreme value for those node pairs with no existing paths.

Table 6 shows the results for DiGT and its variant, DiGT-inf. We see that on dense graphs like `Twitter`, the accuracy difference between these two methods is small (e.g., at most 1.57% on the `Twitter5`). In contrast, on sparse graphs such as `FlowGraph` and `Malnet-sub`, the DiGT-inf variant results in a significant decline in performance (e.g., a 5.69% drop on `Malnet-sub`).

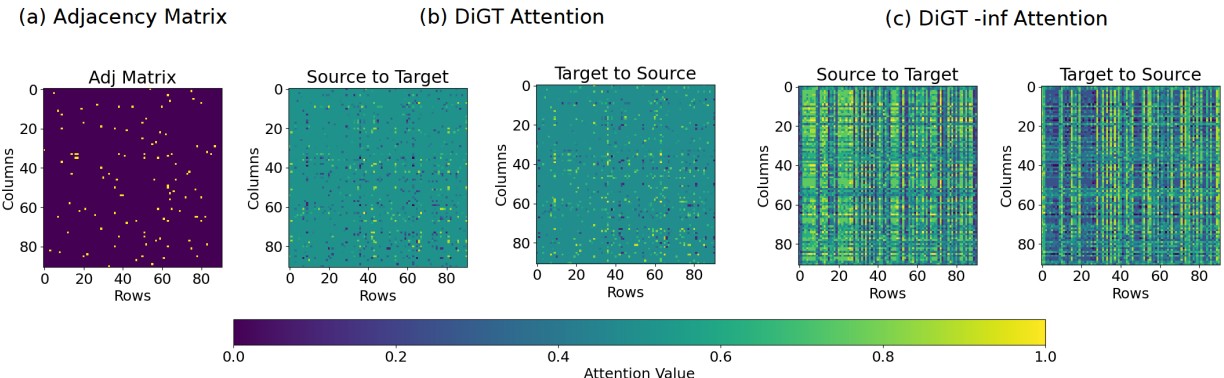

Figure 5: DiGT and DiGT-inf attention matrices for a selected `FlowGraph6` graph.

Table 6: *Ablation:* $\mathbf{D}^{(k)}$

| Model | FlowGraph2 | FlowGraph3 | FlowGraph6 | Twitter3 | Twitter5 | Malnet-sub | MNIST | CIFAR10 |
|---|---|---|---|---|---|---|---|---|
| DiGT | **98.00 +/- 0.54** | **74.61 +/- 1.95** | **46.03 +/- 0.41** | **93.33 +/- 0.64** | **86.67 +/- 0.52** | **80.78 +/- 1.76** | **98.02 +/- 0.10** | **67.05 +/- 0.20** |
| DiGT-inf | 92.08 +/- 0.59 | 69.22 +/- 0.91 | 37.53 +/- 1.26 | 91.17 +/- 0.85 | 84.10 +/- 1.67 | 75.09 +/- 1.09 | 93.52 +/- 0.95 | 49.34 +/- 0.73 |

To better understand the results, Table 7 presents an example of attention values, for a pair of nodes, before and after applying the softmax along the stack dimension in our directional attention approach. For an arbitrary node pair, there are three possible relationships: unconnected, bidirectional, or unidirectional. Both models exhibit identical re-

Table 7: *Attention Values Applying softmax*

| Relationship | Before softmax | | After softmax | |
|---|---|---|---|---|
| | DiGT | DiGT-inf | DiGT | DiGT-inf |
| Unconnected | 0, 0 | $-\infty, -\infty$ | 0.5,0.5 | 0.5,0.5 |
| Bi-directional | 0.7,0.4 | 0.7,0.4 | 0.57, 0.43 | 0.57, 0.43 |
| Uni-directional | 0.7,0 | $0.7, -\infty$ | 0.67, 0.33 | 1.0, 0.0 |

sults for pairs that are either unconnected or bi-directional. The divergence occurs with uni-directional connection: DiGT assigns an extreme value of $-\infty$, whereas DiGT uses 0. Consequently, DiGT-inf yields a pair $(1.0, 0.0)$ making the attention values solely dependent on one direction (in this case, from source to target). Figure 5 illustrates this effect. In other words, for unidirectional node pairs, a hard thresholding using $-\infty$ blocks information flow completely in one direction (with the non-existent edge), whereas using soft thresholding of 0 still favors the direction that exists, but also allows flow in the opposite direction, which leads to SOTA performance of DiGT.

## 5 Conclusions and Future Work

In this paper, we present DiGT, a novel architecture for capturing graph directionality using transformers. We empirically evaluate its classification accuracy on real and synthetic graph datasets and demonstrate its performance gains against state-of-the-art GTs and GNNs. We conduct several ablation studies to reveal the need for a balanced treatment between local and global attention in node and edge encoding channels. Our experiments promote the view that GNNs and GTs could be complementary with the shortcomings of the former (over-smoothing, over-squashing, and limited expressiveness) mitigated by the strengths of the latter and vice versa (with scalability limitations, non-standard graph encodings for attention bias, as potential pitfalls in GTs).

In the future, to overcome the limitations of moderate-sized graph data that are imposed by the quadratic attention complexity, i.e., to scale up the attention mechanism, we plan to explore recently proposed approaches to expand the context (Bertsch et al., 2023; Tay et al., 2022), and study their effectiveness for directed graph datasets.

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

## Appendix

## A    Experiment details

We test our code on a node with NVIDIA V100 GPUs (32GB RAM), 20-core 2.5Ghz Intel Xeon CPU (768GB RAM), running Linux. We use Python and specifically the PyTorch library for our implementation. Our code is available via Github: https://github.com/Qitong-Wang/Directed-Graph-Transformers.

For a fair comparison, we run all models with the number of learnable parameters, around $100K$ with 4 layers for the graph classification tasks. We fix the batch size to 32, the number of maximum epochs to 200, and we employ grid search for tuning the learning rate $\eta \in \{2^i \times u \mid i = 0, 1, 2, 3, 4\}$ with $u = 5 \times 10^{-4}$, choosing $\eta = 5 \times 10^{-4}$ (or $i = 1$) for EGT and DiGT models on `MNIST` and `CIFAR10` datasets, and $\eta = 8 \times 10^{-3}$ (or $i = 4$) for all other datasets. We employ a grid search for tuning the number of hops $k \in \{1, 3, 5, 10, 25\}$; we also employ a grid search for normalization [no normalization, batch normalization, layer normalization]. We apply a similar process for other models. Table 8 contains parameters used for DiGT.

We compare DiGT with GCN (Kipf & Welling, 2016), GAT (Veličković et al., 2017), and PNA (Corso et al., 2020) as the representatives of GNNs, and with GT (Dwivedi & Bresson, 2021), SAN (Kreuzer et al., 2021), EGT (Hussain et al., 2022), Exphormer (Shirzad et al., 2023), and Digraph-Transformer (Geisler et al., 2023) as the representatives of GTs. We reimplemented Digraph Transformer (Digraph-T) since we were not able to directly run their github code, despite a lot of effort to debug and fix the problems encountered. We followed the same approach as done in  (Geisler et al., 2023) for Digraph-T. That is, we modified the SAT implementation (Chen et al., 2022) (in Pytorch) and added the Magnetic Laplacian position encodings. Furthermore, as suggested, we also modified the SAT framework by replacing the GCN layer with three bidirectional GNN layers, using GELU instead of ReLU for the activation function, and adding dropout on the node features. We did not modify the softmax function since our datasets do not suffer from the problem of class imbalance of the special tokens.

We train all methods on `FlowGraph`, `Twitter`, `Malnet-sub`, and `MNIST/CIFAR10`, respectively. To conduct a fair comparison, if the models provide a parameter setting for a similar dataset (e.g., Exphormer has a parameter setting for `Malnet-tiny`), we will keep the same settings and shrink the dimension of hidden layers. If the models do not provide a parameter setting for a dataset, we follow the settings of DiGT, and adjust the hidden dimension. For PNA, we tried grid search over {identity, amplification, attenuation} for the scalers and {mean, max, std, var, sum} for the aggregators.

Table 8: *Training Settings (default values).*

| Hyperparameters | FlowGraph Twitter | MNIST CIFAR | MALNETsub |
|---|---|---|---|
| Batch Size | 32 | 32 | 16 |
| Number of Epochs | 200 | 200 | 200 |
| Early Stops | 0 | 10 | 0 |
| Max Learning Rate ($\eta$) | 0.008 | 0.0005 | 0.008 |
| Number of Virtual Nodes ($q$) | 0 | 0 | 0 |
| Number of Layers | 4 | 4 | 3 |
| Number of Heads ($h = d/d_p$) | 8 | 8 | 4 |
| Node dimensionality ($d$) | 32 | 32 | 44 |
| Edge dimensionality ($d_e$) | 32 | 32 | 32 |
| PE dimension ($r$) | 25 | 8 | 25 |
| Batch normalization | True | False | True |
| Layer normalization | False | True | False |
| Number of Parameters | 93,450 for `FlowGraph2` | 91,538 for `MNIST` | 102,403 |

# B  Dataset details

Table 9 shows the dataset statistics for each of the datasets used in our experiments.

Table 9: *Datasets details: Number of graph instances G in the dataset, average number of nodes $|N|$, average number of directed edges $|E|$ and the number of graph classes $n_c$ are tabulated.*

| Dataset | | $G$ | Avg. $|N|$ | Avg. $|E|$ | $n_c$ |
|---|---|---|---|---|---|
| FlowGraph2 | | 2000 | 114.46 | 150.30 | 2 |
| FlowGraph3 | | 3000 | 111.66 | 150.70 | 3 |
| FlowGraph6 | | 6000 | 111.40 | 149.93 | 6 |
| Twitter3 | | 2919 | 131.76 | 2237.55 | 3 |
| Twitter5 | | 4865 | 131.76 | 2208.79 | 5 |
| MNIST | | 70000 | 70.57 | 564.53 | 10 |
| CIFAR10 | | 60000 | 117.63 | 941.07 | 10 |
| Malnet-sub: | Train | 1434 | 84.85 | 126.14 | 5 |
| Malnet-sub: | Val/Test | 1010 | 466.72 | 938.69 | 5 |

## B.1  `Twitter` Datasets

We use 973 directed ego-networks from Twitter[4], each corresponding to some user $u$ (*ego*): the ego-network is between $u$'s friends also referred to as *alters* (Leskovec & Mcauley, 2012). If nodes $v_i$, $v_j$ are in $u$'s ego-network then $u$ follows them and if $v_i$ follows $v_j$ then there is a directed edge $v_i \mapsto v_j$ in the ego-network. We introduce perturbations to each of these real ego-networks where a perturbation can be either (i) *rewiring* of an existing edge (an $(a, b) \in E(ego(u))$, where it is deleted and replaced by an edge $(c, d)$ where nodes $c, d$ are randomly selected from $V(ego(u))$), or (ii) *reversing* of the direction of an existing edge $(a, b) \in E(ego(u))$, where it is replaced by $(b, a)$. The percentage of the perturbed edges in an ego-network can be $[0, 25, 50, 75, 100]\%$. Rewiring and reversing the direction of edges takes place with equal probabilities. So, for each of the percentages, 973 new perturbed ego-networks are generated, each labeled with the corresponding perturbation percentage. We refer to the collection of the $5 \times 973$ perturbed `Twitter` datasets as `Twitter5` (5 labels/classes). Similarly, if we get $3 \times 973$ of them corresponding to perturbation percentages $[0\%, 50\%, 100\%]$, then we have the `Twitter3` dataset (3 labels/classes).

## B.2  `FlowGraph` Datasets

Given the limitations of existing benchmarks, we introduce a family of directed graph datasets that explicitly relate the edge direction pattern in graphs to their classification labels. In particular, we generate graphs with their nodes organized in successive layers and then we leverage the notion of a flow between the layers through directed edges: for a predefined subset of layers, graphs with different aggregate flow between successive layers in the subset are assigned different labels. Our `FlowGraph` generator is modeled after the *Directed Stochastic Block Model (DSBM)* (Malliaros & Vazirgiannis, 2013). Following the notation in (He et al., 2021), we organize $N$ graph nodes into $K$ clusters and define cluster adjacencies in a meta-graph adjacency matrix $\mathbf{F}$, with its entries $\mathbf{F}_{kl}$ marking the allowance of directed edges from nodes in cluster $k$ to those of cluster $l$. More specifically, we assume that the node clusters are arranged sequentially, $l = 0, 1, \ldots, K - 1$ (say from left to right) and a subset of its first $l_S < K$ consecutive clusters define a subgraph $S$. In `FlowGraph` we allow directed edges between nodes belonging to all clusters with the probability being a small noise parameter $\eta$ (typically $\eta = 0.01$). Then for directed edges between nodes in successive clusters, with the source node $l$ being in a cluster in subgraph $S$, we set $F_{l,l+1}$ to a percentage $f\%$. These percentages are different for different classes and depend on the number of classes $n_c$. In our experiments, for all generated graphs we set $N = 150$, $K = 10$, $l_S = 4$. We generate 3 graph datasets: one dataset for each of the $n_c = 2, 3, 6$-class cases.

---

[4]https://snap.stanford.edu/data/ego-Twitter.html

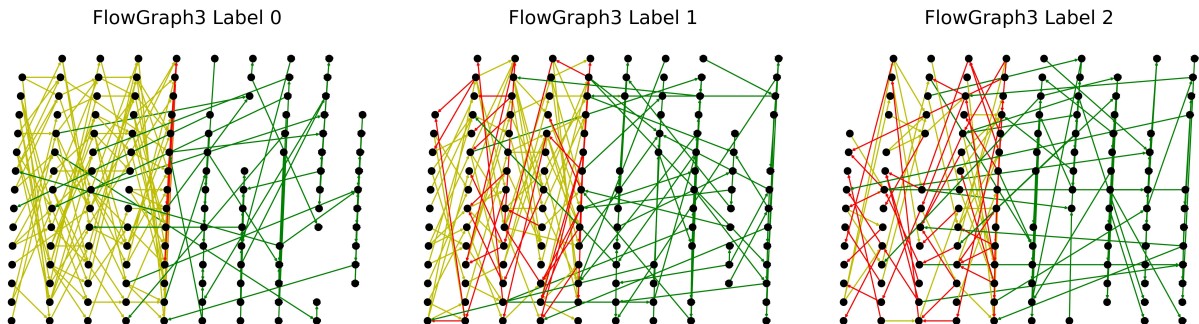

Figure 6: The visualization of three categories of graphs in `FlowGraph3` dataset.

Figure 6 visualizes `FlowGraph3`. The yellow edges are the flows from left to right, the red edges are the flows from right to left, and the green edges are noises. In all three samples, half of the edges are noise, shown with green edges. In the leftmost sample, besides the noise edges, almost all the edges have the flow from left to right; in the middle sample, 75% of the edges have the flow from left to right; whereas in the right sample, 50% of the edges have the flow from left to right.

## C  Flipping Edge Directions

We reverse the direction of 25% and 50% of graph edges, randomly selected, and empirically evaluate the importance of directionality in all the datasets. Table 10 lists our findings. We use GAT(Veličković et al., 2017) and EGT (Hussain et al., 2022) as the representatives for graph neural networks and graph transformers. We confirm that the *derived* notion of *edge direction* in `MNIST` (Achanta et al., 2012) and `CIFAR10` (Krizhevsky et al., 2009) is not significant: classification results from both EGT and GAT models are almost agnostic to edge direction flips in these datasets; the differences between the vanilla datasets and flipped datasets are at most 2.1%.

For the `Ogbg-Code2` dataset (Hu et al., 2020), we select SAT (Chen et al., 2022) and DAGformer (Luo, 2022), the two top-performing models, as the baseline for testing random flips. We observe that DAGformer only exhibits a 1.2% decrease in performance when 50% of the edges in this dataset are randomly flipped. This suggests that directionality is not a significant factor in this dataset.

We employ the current state-of-the-art model, Exphormer(Shirzad et al., 2023), to examine the significance of direction in `Malnet-tiny` dataset (Freitas et al., 2020). It is observed that there's a 6.12% decrease in performance in the flip50 case. This gap refers to the importance of directionality in the `Malnet-tiny` dataset.

For `FlowGraph` and `Twitter` datasets, the labels are determined by the percentage of edge perturbations. Consequently, the directionality of these graphs plays a critical role. In particular, DiGT accuracy *consistently* decreases across all `FlowGraph` and `Twitter` datasets under all edge reversal percentages. In `FlowGraph`, accuracy drops are sharper (and saturate as we increase the reversal percentage): with 25% of edges flipped accuracy decreases in the range of 46.25% to 28.94%: from 97.42% to 51.17% in `FlowGraph2` and from 46.80% to 17.86% in `FlowGraph6`.

## D  Ablation: Positional Embeddings

We select two techniques for the positional encodings: Singular Value Decomposition (SVD), and the Magnetic Laplacian (Geisler et al., 2023). Here are the details for these two methods:

**Singular Value Decomposition**  Given an adjacency matrix $\mathbb{A}$, we apply Singular Value Decomposition (SVD) to decompose it into $\mathbb{A} = \mathbf{U}\Sigma\mathbf{V}^*$ where $\mathbf{U}$ and $\mathbf{V}^*$ are orthogonal matrices and $\Sigma$ is a diagonal matrix

Table 10: *Random Flips of `MNIST`, `CIFAR10`, `Ogbg-Code2`, `Malnet-tiny` datasets.*

| Model | MNIST | CIFAR10 | Model | Ogbg-Code2 | Model | Malnet-tiny | Malnet-sub |
|---|---|---|---|---|---|---|---|
| GAT | 95.54 +/- 0.21 | 64.22 +/- 0.46 | SAT | 19.37 +/- 0.03 | Exphormer | 94.02 +/- 0.21 | 79.71 +/- 0.37 |
| GAT-Flip25 | 93.92 +/- 0.22 | 62.86 +/- 0.18 | SAT-Flip25 | 18.72 +/- 0.08 | Exphormer-Flip25 | 88.77 +/- 0.41 | 70.93 +/- 0.73 |
| GAT-Flip50 | 93.43 +/- 0.23 | 62.11 +/- 0.78 | SAT-Flip50 | 18.70 +/- 0.03 | Exphormer-Flip50 | 87.90 +/- 1.65 | 71.51 +/- 0.45 |
| EGT | 98.17 +/- 0.09 | 68.70 +/- 0.41 | DAG | 20.2 +/- 0.2 | | | |
| EGT-Flip25 | 97.90 +/- 0.11 | 67.27 +/- 0.56 | DAG-Flip25 | 18.9 +/- 0.2 | | | |
| EGT-Flip50 | 97.99 +/- 0.09 | 67.28 +/- 0.38 | DAG-Flip50 | 19.0 +/- 0.1 | | | |

Table 11: *Random Flips of `FlowGraph` and `Twitter` datasets.*

| Model | FlowGraph2 | FlowGraph3 | FlowGraph6 | Twitter3 | Twitter5 |
|---|---|---|---|---|---|
| GAT | 84.92 +/- 1.90 | 58.83 +/- 1.47 | 30.31 +/- 0.28 | 74.59 +/- 1.59 | 56.79 +/- 0.05 |
| GAT-Flip25 | 73.67 +/- 0.72 | 44.16 +/- 1.70 | 48.62 +/- 0.80 | 67.69 +/- 1.06 | 48.62 +/- 0.80 |
| GAT-Flip50 | 51.00 +/- 3.89 | 33.00 +/- 2.38 | 17.31 +/- 0.66 | 65.64 +/- 1.71 | 44.34 +/- 2.05 |
| DiGT | 98.00 +/- 0.54 | 74.61 +/- 1.95 | 46.03 +/- 0.41 | 93.33 +/- 0.64 | 86.67 +/- 0.52 |
| DiGT-Flip25 | 51.17 +/- 1.25 | 33.50 +/- 0.36 | 17.86 +/- 0.92 | 89.29 +/- 0.90 | 77.47 +/- 0.90 |
| DiGT-Flip50 | 49.67 +/- 1.39 | 32.33 +/- 1.66 | 16.78 +/- 0.04 | 82.96 +/- 1.13 | 65.44 +/- 0.38 |

containing singular values. After sorting the singular values in $\Sigma$, we select the top $p$ columns (rows) from both $\mathbf{U}$ and $\mathbf{V}^*$, where $p$ represents the dimensionality of our positional encodings. We utilize them as the dual positional encodings in our model for our input layer as follows where $\mathbf{N}_f$ is input node features:

$$\mathbf{S} = L_s(\mathbf{U}) + L_f(\mathbf{N}_f) \qquad\qquad \mathbf{T} = L_t(\mathbf{V}^*) + L_f(\mathbf{N}_f) \qquad (11)$$

**Magnetic Laplacian** We first compute a symmetric adjacency matrix $\mathbb{A}_S$ given a directed adjacency matrix $\mathbb{A}$ as $\mathbb{A}_S = \mathbb{A} \vee \mathbb{A}^T$, where $\vee$ denotes the 'or' operation. Then, we compute the symmetric diagonal degree matrix $\mathbf{D}_S$ from $\mathbb{A}_S$. According to (Geisler et al., 2023) and (Furutani et al., 2020), we compute the magnetic Laplacian as

$$\mathbf{L} = \mathbf{D}_S - \mathbb{A}_S \odot \exp(i\Theta^q) \qquad (12)$$

where $\odot$ is Hadamard product, exp is element-wise exponential, $i = \sqrt{-1}$, $\Theta = 2\pi q(\mathbb{A}_{u,v} - \mathbb{A}_{v,u})$, and $q \geq 0$. $\mathbf{L}$ is a Hermitian matrix, and we can get a complex eigenvector with real and imaginary $\mathbf{R}$ and $\mathbf{M}$ components. Then, we select $p$ columns from $\mathbf{R}$ and $\mathbf{M}$. Since we have two pairs of eigenvectors for one node, we change our input layer as the following formula:

$$\mathbf{S} = L_{s1}(\mathbf{R}) + L_{s2}(\mathbf{M}) + L_f(\mathbf{N}_f) \qquad\qquad \mathbf{T} = L_{t1}(\mathbf{R}) + L_{t2}(\mathbf{M}) + L_f(\mathbf{N}_f) \qquad (13)$$

Table 12: *Ablation: Positional Embeddings .*

| Model | FlowGraph2 | FlowGraph3 | FlowGraph6 | Twitter3 | Twitter5 | Malnet-sub | MNIST | CIFAR10 |
|---|---|---|---|---|---|---|---|---|
| DiGT-Magnet | **98.00 +/- 0.54** | **74.61 +/- 1.95** | **46.03 +/- 0.41** | **93.33 +/- 0.64** | **86.67 +/- 0.52** | **80.78 +/- 1.76** | 97.25 +/- 0.22 | 63.10 +/- 0.94 |
| DiGT-SVD | 97.42 +/- 0.82 | 74.55 +/- 0.69 | 45.83 +/- 0.25 | 91.67 +/- 0.79 | 85.94 +/- 0.25 | 77.12 +/- 3.68 | **98.02 +/- 0.10** | **67.05 +/- 0.20** |
| DiGT-No PE | 96.83 +/- 0.47 | 74.83 +/- 1.98 | 45.11 +/- 0.75 | 91.28 +/- 0.98 | 85.88 +/- 0.84 | 80.93 +/- 3.97 | 97.63 +/- 0.10 | 65.55 +/- 1.35 |

Table 12 presents the comparative results of these strategies, compared with having no positional encodings. We see that Magnetic Laplacian encodings exhibit superior performance in datasets where directionality is a critical factor, such as `FlowGraph`, `Twitter`, and `Malnet-sub`. Conversely, SVD is more effective in scenarios where directionality is less relevant, such as `MNIST` and `CIFAR10` datasets. Consequently, DiGT follows the same strategy to select the most suitable positional encoding method. Furthermore, the table suggests that incorporating positional encodings generally enhances learning performance, with the exception of applying SVD to the `Malnet-sub` dataset.

## E  Ablation: Virtual Nodes

We investigate the virtual node method from (Hussain et al., 2022): adding $q$ virtual nodes and establishing bidirectional edges between each node from the original graph and each virtual node. In the output layer, it aggregates the output embedding by averaging the node embeddings exclusively from the virtual nodes instead of from all nodes. However, we find that for directed graphs, it is more effective to add edges from the graph to the virtual nodes only, rather than introducing bidirectional connections. Table 13 summarizes our findings for a different number of virtual nodes, $q \in \{1, 3, 5\}$, denoted as DiGT-VN$q$. Note that DiGT by default does not use virtual nodes ($q = 0$). We observe that for `FlowGraph3` dataset, DiGT-VN3 outperforms DiGT by 1.32%. For `FlowGraph6` dataset, DiGT-VN1 shows a 0.72% improvement over DiGT. However, these differences are slight. Additionally, in other datasets, the introduction of virtual nodes tends to decrease performance. Consequently, we decide not to incorporate virtual nodes in our default model due to its limited benefits and potential drawbacks.

Table 13: *Ablation: Virtual Nodes.*

| Model | FlowGraph2 | FlowGraph3 | FlowGraph6 | Twitter3 | Twitter5 | Malnet-sub | MNIST | CIFAR10 |
|---|---|---|---|---|---|---|---|---|
| DiGT | **98.00 +/- 0.54** | 74.61 +/- 1.95 | 46.03 +/- 0.41 | **93.33 +/- 0.64** | **86.67 +/- 0.52** | **80.78 +/- 1.76** | **98.02 +/- 0.10** | **67.05 +/- 0.20** |
| DiGT-VN1 | 97.25 +/- 0.41 | 75.28 +/- 0.44 | **46.75 +/- 1.76** | 92.02 +/- 0.29 | 86.63 +/- 0.64 | 75.93 +/- 0.59 | 97.24 +/- 0.12 | 64.15 +/- 0.46 |
| DiGT-VN3 | 97.83 +/- 0.51 | **75.94 +/- 0.70** | 45.75 +/- 1.49 | 92.82 +/- 0.14 | 86.32 +/- 0.86 | 78.97 +/- 0.82 | 97.32 +/- 0.29 | 64.83 +/- 0.76 |
| DiGT-VN5 | **98.00 +/- 0.35** | 75.89 +/- 1.76 | 46.14 +/- 1.30 | 91.05 +/- 0.32 | 86.63 +/- 0.49 | 76.75 +/- 0.52 | 96.96 +/- 0.13 | 64.63 +/- 0.69 |

## F  Ablation: Undirected Graph Datasets

Table 14: *Ablation: Undirected Graph Datasets.*

| Model | Zinc 100K | Zinc 500K |
|---|---|---|
| GCN (Kipf & Welling, 2016) | 0.459 +/- 0.006 | 0.367 +/- 0.011 |
| GAT (Veličković et al., 2017) | 0.475 +/- 0.007 | 0.384 +/- 0.007 |
| GatedGCN  (Bresson & Laurent, 2017) | 0.375 +/- 0.003 | 0.282 +/- 0.015 |
| GraphSage (Hamilton et al., 2017) | 0.468 +/- 0.003 | 0.398 +/- 0.002 |
| PNA (Corso et al., 2020) | 0.188 +/- 0.004 | 0.142 +/- 0.010 |
| EGT  (Hussain et al., 2022) | 0.277 +/- 0.019 | 0.228 +/- 0.020 |
| EGT+DO  (Hussain et al., 2022) | **0.143 +/- 0.011** | **0.108 +/- 0.009** |
| DiGT | 0.213 +/- 0.031 | 0.134 +/- 0.039 |

While undirected graphs are not our primary focus, we aim to demonstrate that DiGT remains highly effective on such datasets. We choose to further evaluate DiGT on the `Zinc` (Irwin & Shoichet, 2005)dataset, which contains 12K undirected graphs, for the graph regression task. Table 14 reports the mean absolute error (MAE; lower is better) for the `Zinc` dataset under two different training settings: approximately 100K parameters and 500K parameters. Notably, DiGT exhibits superior performance over GCN, GAT, GatedGCN, GraphSage, and EGT. PNA leverages multiple aggregators with degree-scalers to aggregate the neighbors' features and captures the graph structure to attain better performance, and EGT-DO integrates both EGT with a secondary task of distance prediction. Nevertheless, the results indicate that DiGT is competitive for tasks based on undirected graph datasets.

## G  Complexity and Limitations

We designed our experiments so that we allocate the same hardware resources (GPU/CPU, amount of memory) for all experiments. We also limit the number of learned weight parameters to be the same for all models, as shown in Table 15 (for `Malnet-sub`). The recorded training timings show that our method is very competitive with graph transformers: Training time for DiGT is 42s/epoch (for `FlowGraph6` dataset) and 57s/epoch

Table 15: *Training Settings* (for `Malnet-sub`).

|  | DiGT | EGT | GCN | Exphormer |
|---|---|---|---|---|
| Number of Layers | 4 | 4 | 4 | 5 |
| Number of Heads ($h = d/d_p$) | 8 | 8 | 8 | 4 |
| Node dimensionality ($d$) | 44 | 60 | 72 | 48 |
| Edge dimensionality ($d_e$) | 32 | 48 | 0 | 48 |
| Number of Parameters | 102,403 | 107,489 | 105,845 | 113,645 |

(for `Twitter3` dataset). In comparison: for EGT (Hussain et al., 2022) (also a transformer-based architecture), the training time is 30s/epoch (for `FlowGraph6` dataset) and 50s/epoch (for `Twitter3` dataset). Training time for GNNs differs significantly based on their architectures. Vanilla GCN is fast, the training time is 2s/epoch (for `FlowGraph6` dataset) and 3s/epoch (for `Twitter3` dataset). In contrast, for PNA (Corso et al., 2020), a GCN-based architecture, the training time is 62s/epoch (for `FlowGraph6` dataset) and 297s/epoch (for `Twitter3` dataset). In general, for $N$ nodes and $d$-dimensional vectors for node representations at each layer, graph transformers learn weight matrices (space complexity) with $O(N^2)$ parameters each, and graph neural networks learn weight matrices with $O(d^2)$ parameters each (with both (graph) transformers and GCN producing $N$, $d$-dimensional representations for the nodes), conducting matrix-matrix multiplications respectively of (time) complexities $O(N^2 d)$ and $O(d^2 N)$. Given that for the graph-level tasks we conduct, the graphs are relatively small (small $N$, comparable or smaller to $d$), this explains the generally favorable performance of our approach. On top of this, transformer architectures like ours, perform multiplications by splitting matrix dimension $d$ into multiple heads and conducting resulting multiplications in parallel (on GPUs).

