# OpenReview forum: "Directed Graph Transformers"
_TMLR — Accepted by TMLR_

### Review · Reviewer_rHPc · 2024-03-16

**Summary Of Contributions:**

This paper presents a graph transformer architecture that takes into account the direction of the edges between graph nodes. The proposed architecture utilizes dual encoding, positional encodings that respect the edge directionality, and k-hop filtered attention.

**Audience:**

Yes

**Claims And Evidence:**

Yes

**Requested Changes:**

1. Compare against more powerful GNN models.
2. Include the performance of DiGT in undirectional datasets.
3. Fix typos and notation in order to improve clarity.

**Strengths And Weaknesses:**

Strengths:

$\textbullet $ The topic under consideration appears to be an interesting research direction, which gains attention recently.

$\textbullet$ The rationale behind each architectural choice is well-explained.

$\textbullet$ The synthetic datasets proposed are intriguing and serve to illustrate the impact of directionality effectively.

Weaknesses:

$\textbullet$ The fact that real-world datasets are not sensitive to edges flips raises concerns about the cases where directionality is mandatory to solve the underlying task.

$\textbullet$ The impact of different positional encodings (PEs) appears to be minimal and may introduce computational overhead.

$\textbullet$ Since there exist GNN models which perform, in general, better than GCN and GAT, comparing DiGT against other state-of-the-art GNN models could provide valuable insights into its performance.

$\textbullet$ Evaluating DiGT's performance on undirectional datasets and comparing its embeddings' consistency with other architectures could enhance understanding.

$\textbullet$ There exist some typos and confusing notation, X is used for input features and for the output of the model after concatenation. Moreover, some tables are too small and make it difficult to be read, e.g. Table 2.

---

> ### Author Response · Authors · 2024-03-30
>
> Thank you for sharing your feedback on our paper. Here are our responses to the weaknesses you highlight:
>
> * 1. The fact that real-world datasets are not sensitive to edges flips raises concerns about the cases where directionality is mandatory to solve the underlying task.*
>
> **Response** The key distinction between undirected and directed graphs is the directionality. To investigate this, we designed experiments across three distinct categories to obtain comprehensive understanding:
>
> a) Synthetic datasets with only random flipping edges (FlowGraph Datasets) : This setup aims to assess whether a model can capture directionality since the only differences among the categories are the percentages of random flipped edges.
>
> b) Real-World Datasets with rewiring and flipping edges (Twitter Datasets): Here we leverage both rewiring of edges and flipping the direction to create "incorrect" edges.
>
> c) Unaltered Real-World Datasets (Malnet-sub).
>
> We also conducted some more tests on undirected datasets by transforming the directed graphs into undirected graphs. We select the FlowGraph2 and Twitter5 datasets as the representatives for our study. The table below summarizes the results, where Flow2-Un and Twitter5-Un denote the undirected versions of the original datasets. We can see clearly that all models struggle to learn on the undirected graphs. Both this experiment and the random flip experiment underscore the importance of directionality in the FlowGraph2 and Twitter5 datasets. This shows that in cases where directionality matters, DiGT does deliver superior performance.
>
> Datasets     | Flow2   |Flow2-Un | Twitter5 |Twitter5-Un |
> |-------------|---------|---------|----------|------------|
> | GCN         | 87.50   | **52.17**   | 61.23    | **48.01**      |
> | GatedGCN    | 93.67   | 50.00   | 62.97    | 47.28      |
> | Exphormer   | 96.72   | 49.25   | 72.72    | 47.71      |
> | DiGT        | **98.00**   | 51.17   | **86.67**    | 47.25
>
> *2. The impact of different positional encodings (PEs) appears to be minimal and may introduce computational overhead.*
>
> **Response** Using more effective PEs does help improve performance, as we can see in Table 11 in Appendix D. In fact, all of the directed datasets benefit from using the Magnetic Laplacian PEs, whereas CIFAR10 and MNIST, where direction is not critical, also benefit from SVD-based PEs. Please note that there is only a small, one-time overhead of computing the PEs, and subsequent runs simply use the pre-computed PEs. The overhead is small since these graphs are medium sized (with an average number of nodes ranging from 70 to 466; see Table 8).
>
>
> *3. Since there exist GNN models which perform, in general, better than GCN and GAT, comparing DiGT against other state-of-the-art GNN models could provide valuable insights into its performance.*
>
> **Response**  PNA is one of the GNN models that obtains state-of-the-art results for Zinc and CIFAR10 datasets, so we selected it as the representative GNN in our study. [1] is a heavily cited (over 800 citations) paper, and for the graph classification task they show that GraphSage and GatedGCN are their two best performing GNN-based models. Following the reviewer's suggestion, we performed experiments with these two additional GNN models on two of our datasets. The results, presented in the table below, highlight that DiGT surpasses the performance of these GNN models.
>
> | | Flow2   | MalnetSub|
> |-------------|---------|----------|
> | GCN         | 87.50   |  78.75   |
> | GatedGCN    | 93.67   |  80.54   |
> | GraphSage   | 96.72   |  79.94   |
> | DiGT        | **98.00**   |  **80.78**   |
>
> [1] Dwivedi, V. P., Joshi, C. K., Luu, A. T., Laurent, T., Bengio, Y., \& Bresson, X. (2023). Benchmarking graph neural networks. Journal of Machine Learning Research, 24(43), 1-48.

---

> ### Author Response · Authors · 2024-03-30
> **Part 2**
>
> *4. Evaluating DiGT's performance on undirectional datasets and comparing its embeddings' consistency with other architectures could enhance understanding.*
>
> **Response** Since CIFAR10 and MNIST are essentially undirected, Table 2 in our paper provides an initial glimpse on how it behaves on undirected datasets. Furthermore, we included in response 1 above the results of several models when we convert the directed datasets like Flowgraph2 and Twitter5 to undirected graphs. We observed there that DiGT is competitive with other GNN/GT hybrids on undirected graphs.
>
> We chose to further evaluate DiGT on the Zinc dataset, which contains 12K undirected graphs, for the graph regression task. The table below shows the MAE (mean absolute error; lower is better) for the Zinc dataset.
> Notably, DiGT exhibits superior performance over GCN, GatedGCN and EGT. PNA leverages multiple aggregators with degree-scalers to aggregate the neighbors' features and captures the graph structure and attains better performance. Nevertheless, the results indicate that DiGT is competitive for this task also.
>
> | Datasets    | Zinc    |
> |-------------|---------|
> | GCN         | 0.459   |
> | GatedGCN    | 0.375   |
> | EGT         | 0.277   |
> | PNA         | **0.188**   |
> | DiGT        | *0.213*   |
>
> *5. There exist some typos and confusing notation, X is used for input features and for the output of the model after concatenation. Moreover, some tables are too small and make it difficult to be read, e.g. Table 2.*
>
> **Response** We will improve the writing and rename these matrices; we will also introduce a table to summarize the key terms used in our submission and
> adjust the font in the tables. Thank you for pointing out these issues.
>
>
> *Requested Changes: Compare against more powerful GNN models.
> Include the performance of DiGT in undirectional datasets.
> Fix typos and notation in order to improve clarity.*
>
> **Response** We have addressed these concerns as described in our response above.

---

### Review · Reviewer_NBzC · 2024-03-19

**Summary Of Contributions:**

In this paper, the authors propose a graph transformer architecture called DiGT, as well as synthetic datasets of graph for whom direction of edges is crucial in labelling the nodes. Based on a dual encoding mechanism, DiGT recomputes a latent directed adjacency matrix (in fact, two) between each nodes (or limited to k-hops of the original graph) at each layer.

**Audience:**

Yes

**Claims And Evidence:**

Yes

**Requested Changes:**

Mostly some questions/precision, see above

**Strengths And Weaknesses:**

Strength:
- an interesting idea on an important topic
- new (synthetic or partially synthetic) datasets
- experiments are well-designed

Weaknesses/questions:
- the presentation of the architecture is sometimes pretty confusing, especially for the GNN community (where we tend to look for "message-passing" equations and such. I'm guessing this is eq (8)).
- some notations tend to be repeated for different objects, eg S and T appear everywhere
- I don't know the role of the unnumbered equation at the bottom of page 4. Is it used somewhere in the architecture? Is such a relationship verified between S and T at some point?
- unless I am missing something, eq (5) does not force the adjacency matrices to have nonnegative weights, wouldn't it be more intuitive? (eg using exponentials as in the original attention/GAT)
- the efficiency of the architecture would be more discussed. Complete graphs are sometimes unreachable. The authors mention limiting to k-hops, but still manipulate complete graphs in the equation, can the overall complexity of the architecture be limited to k-hop adjacency matrices if desired? Moreover, the authors propose to learn a new adjacency matrix at each layer, in graph learning it is more usual to learn one matrix for the whole architecture, would such weight tying degrade
- The original graph only intervenes in the input S_r and T_r? (and through eventual k-hop filtering) How are they computed? What makes them particularly "source" or "target"? If the original graph is symmetric, is the whole architecture producing symmetric adjacency matrices?
- is the whole architecture permutation-invariant/equivariant, as this is required of all GNNs? (I think so, but might worth a check)
- just to verify, in the experiments, the architectures you are comparing against are indeed using the oriented graph in the message passing? (such that they would be able to exploit the directionality but are not powerful enough to do so compared to yours)

---

> ### Author Response · Authors · 2024-03-30
>
> Thank you for sharing your feedback. DiGT enhances the ability of graph transformers to discern directional relationships. Graph Neural Networks (GNNs) update model weights by aggregating neighboring node features. Conversely, Graph Transformers (GTs) assume that every node is potentially connected to all other nodes, leveraging an attention matrix to compare the significance of each node pair. It uses a softmax function to emphasize crucial connections. For GTs, the adjacency matrix is an input feature to help the model learn the structure of the graphs. In our submission, we denote the adjacency matrix as $\textbf{A}$ and the attention matrix as $\bar{\mathbf{A}}$; however, we do agree that this distinction may not be clear. In our revised submission, we will rename these matrices and summarize our notation in tabular form.
>
> Below we list our responses to your specific queries:
>
> *1. the presentation of the architecture is sometimes pretty confusing, especially for the GNN community (where we tend to look for "message-passing" equations and such. I'm guessing this is eq (8)).*
>
> *2. some notations tend to be repeated for different objects, eg S and T appear everywhere*
>
> **Response** We will add more explanations, clean up the notation, and add a table of notations/symbols. As for eq (8), it refers to the updated node representations based on the attention applied to the value matrices, so it is equivalent to the message-passing node update in GNNs. Also, S and T denote source and target versions for the various elements in DiGT, ST denotes source-to-target flow, and TS denotes target-to-source flows. Again, we believe a table of notations will help clarify any confusion.
>
> *3. I don't know the role of the unnumbered equation at the bottom of page 4. Is it used somewhere in the architecture? Is such a relationship verified between S and T at some point?*
>
> **Response** The unnumbered equation at the bottom of page 4 provides insight into the HITS algorithm, illustrating how source and target embeddings aggregate information from neighboring nodes. They can be considered as the aggregation equations for a very simple GNN that computes the hubs and authorities scores. We use that as a starting point to build the case for an attention-based aggregation for both source-to-target and target-to-source updates as done in the DiGT architecture. In other words, our approach generalizes that concept.
>
> *4. unless I am missing something, eq (5) does not force the adjacency matrices to have nonnegative weights, wouldn't it be more intuitive? (eg using exponentials as in the original attention/GAT)*
>
> **Response** In Equation 5, $\bar{\mathbf{A}}$ represents the attention matrix before the softmax operation. A negative weight at this stage simply indicates that the relationship between the node pair is less significant. Then, in Equation 7, we use the softmax to convert the raw scores into the non-negative attention scores (as done in attention-based models, but along the directionality axis).

---

> ### Author Response · Authors · 2024-03-30
> **Part 2**
>
> *5. the efficiency of the architecture would be more discussed. Complete graphs are sometimes unreachable. The authors mention limiting to k-hops, but still manipulate complete graphs in the equation, can the overall complexity of the architecture be limited to k-hop adjacency matrices if desired? Moreover, the authors propose to learn a new adjacency matrix at each layer, in graph learning it is more usual to learn one matrix for the whole architecture, would such weight tying degrade*
>
>
> **Response**
> Like all transformers, every node in the context pays attention to every other node. Here our context is the entire graph, and thus we get the standard $O(N^2)$ complexity for the attention matrices. This also means that DiGT, and all regular GT methods, can handle medium-sized graphs, e.g., we have tested on graphs with upto 500 nodes. For larger context one would have to employ other techniques like linearization, or other forms of sparse attention. But those are orthogonal to our contributions, which is to handle directionality effectively in ``dense'' transformers.
>
> Note further that each graph exhibits unique sparsity patterns, which makes it challenging to limit attention to any specific subgraph. What DiGT does enable is the flow of information to any nodes with the k-hops of a given node, but this is a dynamic (though sparse) pattern that depends on the adjacency matrix, and therefore the whole matrix has to be considered. Nevertheless, we conclusively show in the ablation study in the appendix 4.3.2, Table 4, that a restricted $k$-hop filter does help empirically.
>
> Pertaining to multiple adjacency matrices, please note that
> instead of relying only on the input graph structure, DiGT learns a new edge representation at each layer, which allows a more dynamic graph structure to evolve at higher layers. Thus, incorporating learnable weights into edge embeddings enhances the model's flexibility and overall learning performance. In fact, the EGT paper [1] has already demonstrated that integrating learnable edge weights can boost performance.
>
> *6. The original graph only intervenes in the input $S_r$ and $T_r$? (and through eventual k-hop filtering) How are they computed? What makes them particularly "source" or "target"? If the original graph is symmetric, is the whole architecture producing symmetric adjacency matrices?*
>
>
> **Response** In Equation 1, $\mathbf{S}_r$ and $\mathbf{T}_r$ represent the positional encodings within the graph. These notations could be confusing, so we plan to rename them to $\mathbf{P}_s$ and $\mathbf{P}_t$ for clarity. We leverage two methods for positional encoding: Singular Value Decomposition (SVD) and Magnetic Laplacian, both of which support asymmetric adjacency matrices (See Appendix D). Furthermore, if the input graph is undirected, its adjacency matrix will be symmetric, but the attention matrices will not be.
>
> *7. is the whole architecture permutation-invariant/equivariant, as this is required of all GNNs? (I think so, but might worth a check)*
>
> **Response** Yes, our model maintains both permutation invariance, i.e.,
> $f(P X, P A P^T) = f(X, A)$ (where $P$ is a permutation matrix, $X$ a matrix with node encodings as rows, and $A$ is the adjacency matrix), and also permutation equivariance, i.e., $f(P X, P A P^T) = P f(X, A)$. In particular, permutation invariance follows from the fact that both our aggregators and pooling layers ($f()$ functions) are invariant to the order of their input vector sequences. Similarly, permutation equivariance holds because relabeling graph nodes simply permutes their representation inputs.
>
> *8. just to verify, in the experiments, the architectures you are comparing against are indeed using the oriented graph in the message passing? (such that they would be able to exploit the directionality but are not powerful enough to do so compared to yours)*
>
> **Response** Yes, to ensure a fair comparison, we use the same datasets across all experiments. All methods used directed graphs as inputs.
>
> [1] Hussain, Md Shamim, Mohammed J. Zaki, and Dharmashankar Subramanian. "Edge-augmented graph transformers: Global self-attention is enough for graphs." arXiv preprint arXiv:2108.03348 3 (2021).

---

### Review · Reviewer_5DUg · 2024-04-14

**Summary Of Contributions:**

The paper introduces Directed Graph Transformer (DiGT), a novel transformer architecture designed to capture the directionality in graph data, which is a critical aspect often overlooked by existing graph transformers and graph neural networks. DiGT utilizes dual encodings for each node to represent its potential role as either a source or target within directed edges. This approach leverages a multi-head directional attention module that considers edge channels as bias, allowing for the dynamic representation of directed graphs.

**Audience:**

Yes

**Claims And Evidence:**

No

**Requested Changes:**

Please kindly refer to the Strengths And Weaknesses.

**Strengths And Weaknesses:**

Strengths:

* The paper addresses an important problem of explicitly accounting for edge directionality in Transformers, which is a challenging and practical task in graph representation learning.
* The use of dual encodings for nodes to represent their roles as sources or targets provides a more accurate and flexible representation of directed edges.
* Empirical evaluations show that DiGT outperforms many existing graph transformer and graph neural network approaches in terms of classification accuracy on both synthetic and real graph datasets.

Weaknesses:
* The proposed method makes a very incremental improvement over [1] (considering the positional encoding of DiGT is borrowed from [1]), which indicates that DiGT is not the first GT architecture that explicitly takes into account graph directionality, and therefore hinders the novelty of this work. The authors are encouraged to include detailed analysis and comparison with [1] to support the significance of DiGT. From my perspective, without a detailed discussion of how DiGT contributes over [1] as well as thorough empirical comparisons with [1], the current claims regarding the contributions could not make sense.

* The need for a balanced treatment between local and global attention in node and edge encoding channels is a kind of well-explored observations for GTs on undirected graphs [2]. Therefore, this observed pattern made by the authors is not new. Could the authors elaborate more on how this need is unique for GT towards directed graphs?

* The design of each component in DiGT seems to be rather intuitive without strong motivations. It would be better for the authors to provide more detailed explanations of the effectiveness of each component design with possible theoretical groundings.



[1] Transformers Meet Directed Graphs, ICML 2023

[2] Recipe for a general, powerful, scalable graph transformer, NeurIPS 2022

---

> ### Author Response · Authors · 2024-04-28
>
> Thank you for sharing your concerns. Here are our responses to the topics you've highlighted:
>
> *1. The proposed method makes a very incremental improvement over [1] (considering the positional encoding of DiGT is borrowed from [1]), which indicates that DiGT is not the first GT architecture that explicitly takes into account graph directionality, and therefore hinders the novelty of this work. The authors are encouraged to include detailed analysis and comparison with [1] to support the significance of DiGT. From my perspective, without a detailed discussion of how DiGT contributes over [1] as well as thorough empirical comparisons with [1], the current claims regarding the contributions could not make sense.*
>
> **Response**
> Thanks for suggesting that we compare with the digraph transformer proposed in [1]. Unfortunately, this is something we tried to do, and despite our many efforts, we have not been successful in running their model. Despite fixing numerous issues related to different package versioning, ultimately the code still does not run, giving errors during the training stage. We had previously also contacted the authors, but we did not receive any meaningful suggestions. For these reasons, we are not able to experimentally compare with their work.
>
> However, it is important to note that our method cannot simply be dismissed as being an extension of [1]. First of all, the magnetic Laplacian was originally proposed in [2]. Second, in our approach, the magnetic Laplacian (ML) encodings make only a small difference, as shown in Table 11 in the Appendix. For example, an accuracy of 46.03 with ML vs 45.83 with SVD for Flowgraph3, or 86.65 with ML vs 85.94 with SVD for Twitter5. The most benefit is for Malnet-sub.  We utilize position encodings (PE) merely as one of the techniques for input node features, which is a minor component of our overall architecture. In contrast, our main contribution is the integration of bidirectional attention for graph transformers.
>
> *2. The need for a balanced treatment between local and global attention in node and edge encoding channels is a kind of well-explored observations for GTs on undirected graphs [2]. Therefore, this observed pattern made by the authors is not new. Could the authors elaborate more on how this need is unique for GT towards directed graphs?*
>
> **Response** A GraphGPS [3] layer comprises a hybrid of MPNN (message-passing graph neural networks) layer and transformer layer. The innovation of GraphGPS includes integrating positional encodings and structural encodings within the MPNN layer and demonstrating how this combination enhances model performance. Its novelty primarily resides in the MPNN component, with no changes made to the transformer architecture. In contrast, our contribution focuses on the integration of bidirectional attention within the transformer layer. As a result, the innovations in our paper differ significantly from those in GraphGPS.
>
> Additionally, we selected Exphormer [4] as the representative of the model which combines MPNN layers and transformer layers. Exphormer builds on the GraphGPS framework, inheriting all its advantages and flexibility, and was shown to outperform GraphGPS. Therefore, comparing our model to Exphormer provides a more convincing analysis, and we show conclusively that DiGT outperforms Exphormer on all datasets except for CIFAR10 and MNIST where direction does not matter.

---

> > ### Author Response · Authors · 2024-04-28
> > **Part 2**
> >
> > *3. The design of each component in DiGT seems to be rather intuitive without strong motivations. It would be better for the authors to provide more detailed explanations of the effectiveness of each component design with possible theoretical groundings.*
> >
> > **Response** Our DiGT model is inspired by the well-known HITS [5] approach for directed graphs using the notion of authority and hub scores (which, are eigenvectors of the $A^T A$ and $A A^T$ matrices), which generalizes the PageRank algorithm. Our key innovation is to generalize the idea via the use of source/target vectors (embeddings) with a learnable adjacency matrix, and to further propose the novel idea of directional attention. Our work is thus based on a solid theoretical foundation that combines ideas from network science with deep learning on graphs.
> >
> > [1] Simon Geisler, Yujia Li, Daniel J Mankowitz, Ali Taylan Cemgil, Stephan Gunnemann, and Cosmin Padu-
> > raru. Transformers meet directed graphs. In International Conference on Machine Learning, 2023.
> >
> > [2] Furutani, S., Shibahara, T., Akiyama, M., Hato, K., and Aida, M. Graph Signal Processing for Directed Graphs Based on the Hermitian Laplacian. ECML/PKDD Conf, 2020.
> >
> > [3] Radislav Rampasek, Michael Galkin, Vijay Prakash Dwivedi, Anh Tuan Luu, Guy Wolf, and Dominique
> > Beaini. Recipe for a general, powerful, scalable graph transformer. Neural Information Processing Systems, 2022.
> >
> > [4] Hamed Shirzad, Ameya Velingker, Balaji Venkatachalam, Danica J Sutherland, and Ali Kemal Sinop. Exphormer: Sparse transformers for graphs. International Conference on Machine Learning, 2023.
> >
> > [5] Jon M Kleinberg. Authoritative sources in a hyperlinked environment. Journal of the ACM, 46(5): 604–632, 1999.

---

### Decision · Action_Editor_HBQL · 2024-06-01

**Recommendation:** Accept with minor revision

**Comment:**

The paper was reviewed by three reviewers. The overall recommendations from the reviewers were mixed, with two reviewers advocating for weak acceptance and another advocating for rejection. The reviewers raised concerns about the clarity of the presentation, e.g., the notation used in the paper is somewhat confusing. The reviewers also complained about the message passing baselines, mainly that the employed baselines are weak. In the response, the authors addressed most of the reviewers' concerns, but the manuscript has not been revised yet to account for all the reviewers' feedback. In my view, the most severe problem with this paper is that the proposed architecture is not empirically compared against [1]. It appears that the authors have made efforts to train the model proposed in [1], but they have not succeeded because of coding issues. Even though this comparison would strengthen the paper, I believe that it is not mandatory for acceptance. I am thus recommending accept with a minor revision, and I request that the final version of the manuscript carefully considers the reviewers' comments, mainly the items listed below:

- The claim that DiGT outperforms existing graph transformers by a large margin needs to either be verified by comparing against [1] or be sufficiently toned down.
- A comparison against stronger message passing models needs to be added to the manuscript.
- The presentation of the paper needs to be improved.
- A detailed discussion needs to be added in the manuscript on how DiGT differs from the method proposed in [1].

**Audience:**

How to take edge direction into account in graph transformer architectures is an important problem. Therefore, the findings of this paper will be of interest to some individuals in TMLR's audience.

**Claims And Evidence:**

This paper proposes DiGT, a graph transformer architecture which can handle directed graphs. The proposed architecture indeed takes edge direction into account and seems reasonable. The paper claims that on datasets that consist of inherently directed graphs, the proposed model outperforms existing graph transformers and message passing models by a large margin. It is not clear whether this claim is actually true since, as discussed by Reviewer 5DUg, the proposed DiGT model is not empirically compared against the one proposed in [1], which is very similar to DiGT. I thus ask the authors to tone down this claim or compare against the model proposed in [1] and update their claim.

[1] Geisler, S., Li, Y., Mankowitz, D. J., Cemgil, A. T., Günnemann, S., & Paduraru, C. Transformers meet directed graphs. In Proceedings of the 40th International Conference on Machine Learning, pp. 11144-11172, 2023.

---

> ### Author Response · Authors · 2024-06-29
>
> Thank you for the valuable comments and suggestions to improve. We have made all required changes as detailed below:
>
> *1. The claim that DiGT outperforms existing graph transformers by a large margin needs to either be verified by comparing against [1] or be sufficiently toned down.*
>
> **Response**  According to Table 1 in [1], their method -- Digraph Transformer (Digraph-T) -- achieves state-of-the-art results on the Ogbg-Code2 dataset by leveraging Magnetic Laplacian positional encodings with the modified SAT[2] graph transformer architecture. We selected this version as the representative implementation of Digraph Transformer, since we were not able to directly run their github code, despite a lot of effort to debug and fix the problems encountered. Since both the Magnetic Laplacian positional encodings and SAT code have PyTorch implementations, we integrated these two mechanisms in our Digraph-T implementation.
>
> Following the instructions in the Empirical Evaluation section (Page 8) and Appendix G in [1], we modified the SAT framework by replacing the GCN with three bidirectional GNN layers, using GELU instead of ReLU for the activation function, and adding dropout on the node features. We did not modify the softmax function since our datasets do not suffer from the problem of class imbalance of the special tokens. We ran Digraph-T in PyTorch across all our datasets and generated comparisons for the final-camera version. We made every effort to ensure a fair experiment.
>
> From Table 3 in our manuscript, it is evident that DiGT outperforms Digraph-T across all datasets. Specifically, DiGT achieves a 2.58% improvement on the FlowGraph2 dataset, a 6.09% improvement on the Twitter3 dataset, and a 7.50% improvement on the MalNet-sub dataset.
>
> *2. A comparison against stronger message-passing models needs to be added to the manuscript.*
>
> **Response** The paper [4] is heavily cited (over 800 citations), and it shows that GraphSage[5] and GatedGCN[6] are their two best-performing GNN-based models. Therefore, we added comparisons with these two models for all the datasets we have in our final camera-ready version. Furthermore, we included comparisons of the Zinc[7] datasets with 100K and 500K parameter settings as representatives for testing undirected graphs, in the Appendix.
>
> *3. The presentation of the paper needs to be improved.*
>
> **Response** We added Table 1 that lists the mathematical notations; this appears at the beginning of the DiGT architecture section. The table contains explanations of commonly used symbols and their dimensions. This should help readers understand our DiGT graph structure more easily. Additionally, we revised the writing in this section to make it more concise and clear. We also reviewed the entire paper and improved the overall writing quality.
>
> *4. A detailed discussion needs to be added in the manuscript on how DiGT differs from the method proposed in [1].*
>
> **Response** We highlighted how our model differs from Digraph Transfomer [1] in the related work section. While they utilize Magnetic Laplacian positional embeddings (PEs), we use PEs merely as one of the techniques for input node features, and which comprise only a minor component of our overall architecture. In contrast, our main contribution is the integration of bidirectional attention for graph transformers.
>
>
>
> [1] Geisler, S., Li, Y., Mankowitz, D. J., Cemgil, A. T., Günnemann, S., & Paduraru, C. Transformers meet directed graphs. In Proceedings of the 40th International Conference on Machine Learning, pp. 11144-11172, 2023.
>
> [2] Chen, Dexiong, Leslie O’Bray, and Karsten Borgwardt. "Structure-aware transformer for graph representation learning." International Conference on Machine Learning. PMLR, 2022.
>
> [3] Hu, Weihua, et al. "Open graph benchmark: Datasets for machine learning on graphs." Advances in neural information processing systems 33 (2020): 22118-22133.
>
> [4] Dwivedi, V. P., Joshi, C. K., Luu, A. T., Laurent, T., Bengio, Y., & Bresson, X. (2023). Benchmarking graph neural networks. Journal of Machine Learning Research, 24(43), 1-48.
>
> [5] Hamilton, Will, Zhitao Ying, and Jure Leskovec. "Inductive representation learning on large graphs." Advances in neural information processing systems 30 (2017).
>
> [6] Bresson, Xavier, and Thomas Laurent. "Residual gated graph convnets." arXiv preprint arXiv:1711.07553 (2017).
>
> [7] Irwin, John J., and Brian K. Shoichet. "ZINC− a free database of commercially available compounds for virtual screening." Journal of chemical information and modeling 45.1 (2005): 177-182.